# Functional connectivity patterns of the Giant Toad *Rhinella horribilis* in anthropogenically modified landscapes

Gerardo J. Soria-Ortiz[1,2]*, Leticia M. Ochoa-Ochoa[3], Juan P. Jaramillo-Correa[1], Íñigo Martínez-Solano[4], Ella Vázquez-Domínguez[1]*

**1** Laboratorio de Genética y Ecología, Departamento de Ecología de La Biodiversidad, Instituto de Ecología, Universidad Nacional Autónoma de México, Ciudad de México, México, **2** Posgrado en Ciencias Biológicas, Universidad Nacional Autónoma de México, Unidad de Posgrado, Ciudad de México, México, **3** Museo de Zoología "Alfonso L. Herrera", Departamento de Biología Evolutiva, Facultad de Ciencias, Universidad Nacional Autónoma de México, Ciudad de México, México, **4** Museo Nacional de Ciencias Naturales (MNCN-CSIC), c/ José Gutiérrez Abascal 2, Madrid, Spain

\* evazquez@ecologia.unam.mx (EVD); gsoria@ecologia.unam.mx (GJSO)

## Abstract

Anthropized environments often fragment native habitats and alter the movement of individuals across the modified landscape mosaic, which is significantly challenging for wild species. Deciphering the environmental factors associated with population genetic patterns in modified habitats is essential to understand functional connectivity and for the conservation of wild populations inhabiting increasingly modified habitats. We used ddRAD-seq genomic data to study the genetic diversity, genetic structure and functional connectivity of the Giant Toad, *Rhinella horribilis* populations across two landscapes with distinct levels of habitat modification. We also applied a landscape genetics approach to identify landscape variables (climatic, vegetation, water bodies, land use) associated with the toad's functional connectivity in both landscapes. Structure analysis between the two landscapes show that they are genetically differentiated given their distinct degree of habitat modification. Within landscapes, our results identified lower genetic diversity, higher genetic structure and lower functional connectivity among *R. horribilis* populations in the landscape with higher habitat modification. Results also demonstrate that structure and functional connectivity are significantly influenced by barriers like rivers and roads. Furthermore, water bodies availability was the most important landscape feature for *R. horribilis* connectivity, whereas vegetation cover, solar radiation and relative humidity also played a significant role. Our study illustrates how landscape features in modified habitats can differently determine genetic diversity and functional connectivity patterns, and highlights the importance of working with often-disregarded common species like the Giant Toad. Prioritizing the management of water bodies in our study

**Data availability statement:** The demultiplexed Illumina reads of the Giant Toad Rhinella horribilis were uploaded to NCBI SRA database under accession number PRJNA1335177. The final assembled and filtered datasets (vcf files) used for the study-wide and the two population-study analyses are shared in the FigShare repository: https://doi.org/10.6084/m9.figshare.29913284. Sampling locations, data per individual, methods for all analyses, and genetic results are available in the main text and in the Supporting Information. The R scripts used in the study were compiled into five sections with the different analyses performed, which are included in the Supplementary Methods pdf file.

**Funding:** LMOO and EVD acknowledge project funding from Consejo Nacional de Humanidades, Ciencias y Tecnologías CONAHCyT (grant # PN 2271). LMOO also acknowledges project funding from Programa de Apoyo a Proyectos de Investigación e Innovación Tecnológica (PAPIIT-DGAPA IN220321).

**Competing interests:** The authors have declared that no competing interests exist.

sites and elsewhere would be essential to sustain amphibian population dynamics, enhancing individual movement and genetic exchange.

## Introduction

Adaptive and non-adaptive evolution is crucial for populations to respond to environmental change. Non-adaptive forces include mutation, genetic drift, gene flow and recombination, which are primarily random, whereas adaptive forces like natural selection depend on the fitness (relative or absolute) of individuals within populations [1,2]. Adaptation to anthropized habitats is key for the persistence of species in such environments [3–5]. The global impact of human actions on ecosystems is rapidly altering their abiotic and biotic characteristics. Some direct effects include changes in the structure and configuration of the landscape matrix (e.g., vegetation cover, water bodies, land uses, soil types) and in local microclimatic conditions (e.g., temperature, solar radiation, humidity). Pollution of soils and water bodies derived from agriculture, livestock and industrial activities are also markedly detrimental [6–8]. These modifications have negative effects on the survival and abundance of many species of flora and fauna and constrain the movement of individuals among the remaining habitat fragments or patches. Reduction in individual dispersal, and the consequent loss of gene flow, often promotes genetic drift and differentiation among populations, with detrimental consequences for the long-term maintenance of genetic diversity [9,10]. Nevertheless, some species can persist in modified areas. Persistence can be facilitated by specific adaptations in their life history, ecology and functional traits that counteract the negative consequences of isolation [3,11–13]. Therefore, it is essential to jointly characterize patterns of functional connectivity and decipher the environmental factors associated with species tolerance and/or facilitation in modified habitats. Such framework facilitates the understanding of the effects of anthropization on genetic patterns in wild populations [14–16].

Functional connectivity refers to the degree to which organisms effectively disperse across a landscape, which depends on the species life history traits and the structure and composition of the landscape [16,17]. In amphibians, some landscape factors often negatively impact functional connectivity, including the loss of vegetation cover, the extent of impervious surfaces, road infrastructure, and other changes in land uses [18,19]. As an example, the Northern Two-lined Salamander (*Eurycea bislineata*) that has very low vagility, and the Spotted Salamander (*Ambystoma maculatum*) that is strictly aquatic, maintain connectivity among populations embedded in urban areas, which is facilitated by the presence of streams and riparian vegetation [18,20]. Although much less studied, different climatic [21] and physicochemical variables of water bodies used as breeding sites [8,18,19] are important in facilitating or limiting gene flow among amphibian populations. Identifying environmental variables that most strongly affect amphibian genetic patterns is therefore essential to understand their functional connectivity in modified landscapes.

The southern tropical region of Mexico harbors significant amphibian diversity, where recent discoveries of new species and of species distribution expansions occur often [22–24]. Regrettably, this region also concentrates many threatened species [25] and has been identified as a priority area for amphibian conservation [26,27]. Landscape-level studies with amphibians in Mexico are rather scarce [13]. One example is the study of the Mexican leaf frog (*Agalychnis dacnicolor*) showing that native tropical forest and agriculture areas facilitated its genetic connectivity, whereas open areas (grasslands, human settlements) hindered it [28]. It is therefore urgent to further our knowledge about the effects of habitat modification by human activities on amphibian communities and to study species with different life-history traits.

One of the most common and abundant amphibian species across different environments in the Neotropics is the Giant Toad *Rhinella horribilis*, which is naturally distributed along the Pacific and Gulf of Mexico coasts in Mexico, and through Central America to northern South America. This species tolerates a broad range of climatic conditions and can breed in water bodies with suboptimal characteristics like excess of potassium and sodium ions [29,30]. Its medium-large body size (100–200 mm total length), terrestrial habits, ability to reproduce in both shallow lotic (stream) and lentic (temporary) water bodies, high dispersal capacity and toxicity are some of the traits that make *R. horribilis* a resilient, successful competitor [31,32]. No population genomics studies have been performed with *R. horribilis*, despite its phylogenetic, ecological, and life history traits that make it an excellent system to evaluate functional connectivity patterns in anthropogenically modified landscapes.

In this study, we aimed to (1) characterize the genetic diversity, genetic structure and functional connectivity of *R. horribilis* populations in two landscapes with distinct levels of habitat modification (high and moderate), and to (2) determine landscape variables (climatic, vegetation, water bodies, land use) associated with functional connectivity in both landscapes. Based on the distinct landscapes studied and on *R. horribilis* life history traits, we respectively predicted (1) the two landscapes to be genetically differentiated; (2) low genetic diversity, high genetic structure and low gene flow (functional connectivity) among *R. horribilis* populations in the landscape with higher habitat modification; and (3) genetic connectivity patterns tightly associated with the presence of water bodies irrespective of landscape.

## Materials and methods

### Field sampling, environmental data and landscape features

We worked in two landscapes in the Sierra Madre del Sur, along the Pacific slope in the state of Oaxaca, southern Mexico. The landscapes, separated by ca. 70 km, have experienced a decrease in vegetation cover mainly due to anthropogenic modification for more than 400 years. Each landscape has a different configuration, amount of vegetation cover, and land uses (Fig 1a-c, Table S1 in S2 File); Landscape 1 (P1O; high modification) has lower vegetation cover (47%), higher fragmentation (56.2; CONTAG index) and 50.64% modified land (pastures and technified agriculture, livestock and bare soil) in comparison with Landscape 2 (P2O; moderate modification) with 54.1% vegetation cover, fragmentation (50.6), and 43.5% modified land [33] (Table S1 in S2 File). Sampling was performed during the rainy season (May to September 2019). We chose eight and seven sampling sites in P1O and P2O, respectively (Fig 1b,c), located on either side of a main road or rural town (urbanized area) and separated by a minimum of 1.5 km and a maximum of 23 km (P1O) and 15 km (P2O) distance. This sampling scheme allowed us to test for isolation by barrier, since roads and highly modified habitats are known to represent significant barriers for the connectivity of amphibian populations [18,34]. We set two 50x2 m (100 m²) transects at each sampling site, which were sampled once each for 1.5–2 hours. We collected toe clips from adult (post-metamorphic) *Rhinella horribilis* individuals; tissue samples were stored in eppendorf tubes with 99% ethanol; all individuals were released at their sampling location. Fieldwork, sampling and animal care were performed in strict adherence to the guidelines for working with amphibians and reptiles [35], approved and following the guidelines of the Ethics Committee Facultad de Ciencias-UNAM (project PN 2271), and with the appropriate scientific collecting permit from Secretaría del Medio Ambiente y Recursos Naturales (SEMARNAT; 09/K4-1472/09/18 to LMOO). We followed AmphibiaWeb (https://amphibiaweb.org/) for taxonomy nomenclature.

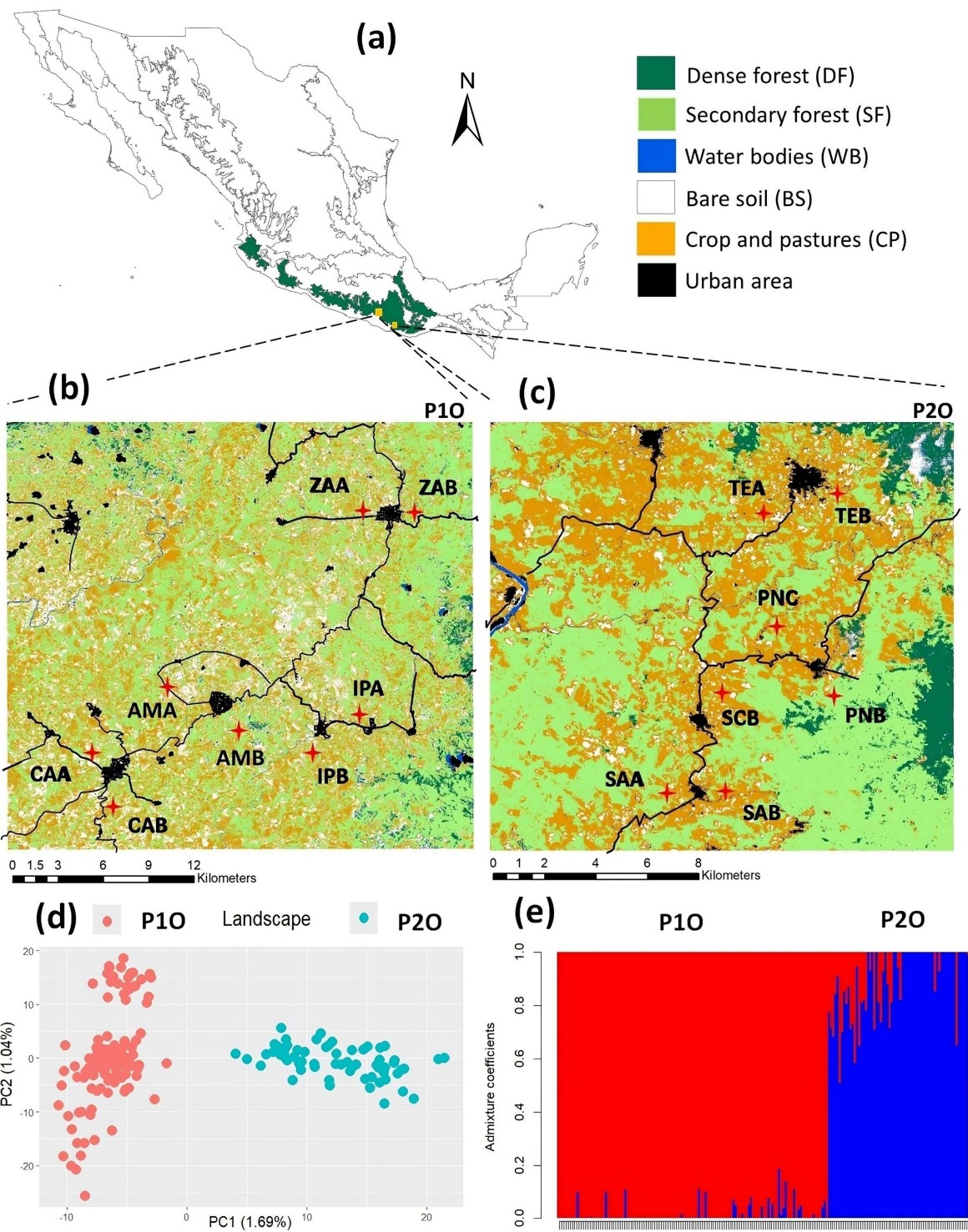

**Fig. 1. Sampling sites for *Rhinella horribilis* within the state of Oaxaca, southern Mexico and landscape structure based on 6034 SNPs.** (a) Sierra Madre del Sur (green outline), (b) landscape 1 (P1O), (c) landscape 2 (P2O). The sampling sites are indicated with red crosses; note they are located on both sides of a road and/or urban town. P1C (b) and P2O (c) have different scales due to distinct extent of study area. (d) Principal components (PCA) and (**e**) admixture coefficient (SNMF) graphs of the 190 individuals from both landscapes (125-P1O; 65 P2O), which are ordered by latitude from left to right. See Table S2 in S2 File for sampling sites names. Information of geostatistical boundaries was obtained from a public domain source [http://geoportal.conabio.gob.mx/#!l=anfibios:1@m=topo] and the map was created by us using QGIS 3.22.10 software.

To characterize the local environment of each landscape, we obtained *in situ* data on climatic and landscape features. For the first, we used a Kestrel 3000 climate meter (Kestrel® Instruments) to take measures of ambient temperature (AT) and relative humidity (RH) along the transects at each sampling site. We also obtained data for the solar radiation (SR) and evapotranspiration (EVA) variables from WorldClim 2.1, at a resolution of 1 km. For the landscape variables, we used the 'Mexican elevational continuum surfaces' v3.0 (resolution of 15 m, available at https://www.inegi.org.mx/app/geo2/elevacionesmex/) to generate the elevation variable (ELV); and extracted information on the main and secondary roads from the National Road Network (http://rnc.imt.mx, downloaded in October 2023) at 10 m resolution; and of rivers and streams (10 m) from the INEGI national hydrology database (https://www.inegi.org.mx, downloaded in October 2023). Finally, we downloaded satellite images (May 2019) from the Landviewer platform (Sentinel-2 L2A) at a 10 m resolution, and estimated indices of normalized vegetation (NDVI), normalized humidity (NDMI) and bare soil (BSI) with QGIS v3.22. We used infrared visualization to classify dense forest, secondary forest, bare soil, crops and pastures, urban areas and temporary water bodies, performing supervised classification by maximum likelihood in the satellite images (see [36]). Based on this classification we built a categorical land use surface (LU). Additionally, we did a kernel density analysis with a 500 m search radius from unique points representing potential temporary water bodies (pixels) and temporal streams (lines). Based on this information, we identified areas with a high probability of presence of temporary water bodies (in the rainy season). For all variables, we built environmental surfaces with QGIS v3.22 [37], which were restricted to the study areas and reclassified to a pixel size of 30x30 m.

## DNA extraction, sequencing, and bioinformatics processing

We extracted genomic DNA from 190 *R. horribilis* individual samples (125 for P1O and 65 of P2O) with the DNeasy Blood and Tissue Kit (Qiagen, Valencia, CA, USA), following the manufacturer's protocol. We confirmed DNA quality and quantity on 1% agarose gels using GelRed® (Biotium, Fremont, CA) and a Qubit™ fluorometer (Invitrogen, Carlsbad, CA). Library preparation and sequencing were performed at the University of Wisconsin Biotechnology Center using the paired-end ddRAD protocol [38] and the restriction enzymes *Pst*I and *Bfa*I. Samples were sequenced on an Illumina NovaSeq lane to obtain 150-bp paired reads.

The total number of reads generated from sequencing was 1,187,625,595. The bioinformatic processing of the raw data comprised the following steps: demultiplexing raw reads with the function *process_radtags* in Stacks v.2.62 [39]; quality filtering with FastQC v.0.11.9 [40] using a Phred quality score <20 and removing adapters, as well as trimming reads to a minimum length of 100 bp with Trimmomatic v.0.36 [41]. We aligned the resulting reads to the *R. marina* reference genome using the function *BWA mem* in BWA v.0.7.1 [42]. Calling of single nucleotide polymorphisms (SNPs) was done with the *ref_map.pl* pipeline and the *populations* module in Stacks, using the parameters p = 7, r = 0.7.

From this first SNPs dataset and considering the objectives of our study, we built three datasets, one that included all the samples from the two landscapes (study-wide dataset) and two more by landscape (population-study-P1O and population-study-P2O). Additional quality filtering was done for each dataset with VCFtools v.0.1.13 [43], SNPfiltR v.1.0.2 [44] in R [45], and *populations* in Stacks. We provide a detailed description of the filtering processes and results in the Supplementary Methods in S3 File. Briefly, we assessed allele balance using *filter_allele_balance* with the default parameters [46]; applied genotype depth and genotype quality using --minDP = 5 and --minGQ = 20, respectively [47]; only biallelic loci were retained; maximum allowed amount of missing data of 20% (*--max-missing*); and a stringent cut-off value of 10kb for the linkage disequilibrium (LD) filter (*--thin*) (see specific details in Supplementary Methods in S3 File). We also removed potential paralogous sequences using a cut-off value of 2x mode, where "x" is per-SNP mean sequencing depth [48], and SNPs with observed heterozygosity >0.50 (*--max-obs-het*). Finally, we assessed a range of minor allele count (*--mac*) values (3–17; see Supplementary Methods in S3 File for all the results), and chose a minor allele count of seven for both the study-wide and the population-study-P1O datasets and of five for the population-study-P2O.

## Genetic structure and diversity patterns

Genetic structure analyses were conducted at two levels, one considering all 15 localities together (study-wide dataset) and the other within each of the P1O and P2O landscapes separately (population-study datasets). We applied two approaches that do not assume an underlying population genetic model, a principal component analysis (PCA) that is not affected by unequal sample sizes [49], and a Discriminant Analysis of Principal Components (DAPC), that uses the *a-score* method to determine the proportion of successful reassignment by individual as a function of the number of retained PCs, which maximizes the differentiation between populations. We applied PCA to both the study-wide and the population-study datasets, while DAPC only for the latter. We ran PCA with the *glpca* function in adegenet v.2.1.3 in R [50], retaining 50 and 26 factors for P1O and P2O, respectively. Miller et al. [51] showed that using *a priori* grouping for DAPC is more efficient to better reflect the underlying $F_{ST}$, even if it is small, and when sampling sizes are small, as it is the case in the two landscapes studied. Therefore, we grouped individuals *a priori* according to the eight and seven sampling sites in P1O and P2O, respectively and ran DAPC with *dapc* in adegenet. The number of retained PCs (P1O=40, P2O=10; S1 Fig in S1 File) was obtained with *xvalDapc* in poppr v.2.9.3 [52]. Additionally, we ran sPCA with *spca* in adegenet for the population-study datasets, a spatial model that is well suited to identify cryptic genetic structure, as it combines the spatial autocorrelation of sampling sites and the genetic variation within sites to maximize the effects of global and local structure [53]. Additionally, we used two methods that are based on sparse non-negative matrix factorization, SNMF [54] and TESS [55]. Both tests compute regularized least-square estimates of admixture proportions to estimate individual ancestry coefficients; TESS further includes the spatial information (geographic coordinates) of each sampling site. We tested SNMF from $K=1$ to $K=3$ with the study-wide dataset; for the population-study we ran SNMF and TESS testing from $K=1$ to the maximum number of populations in each landscape ($K=8$ in P1O and $K=7$ in P2O) with 20 replicate runs per *K-value*, using the function *snmf* in LEA v.3.2.0 [54] and *TESS3* in Tess3r v.1.1.0 [55], both in R. Given that P1O and P2O are clearly genetically separated (see Results), genetic diversity, measured as observed ($H_o$) and expected ($H_e$) heterozygosity, as well as $F_{IS}$, were estimated per landscape with *populations* in Stacks. Statistical differences between landscapes were evaluated using a linear model, considering spatial correlation structure (*corGaus*), genetic diversity as the response variable, and landscape as the explanatory variable, using *gls* in nlme v.3.1 (https://CRAN.R-project.org/package=nlme). We also estimated the genetic differentiation between P1O and P2O with the pairwise fixation index ($F_{ST}$) with *pairwise.neifst* with hierfstat v.0.5–11 [56] in R.

## Landscape genetics analyses

The landscape genetics assessment was done within each P1O and P2O. We performed correlation analyses, based on the surfaces we built for the different environmental variables per landscape, using function *cor* with stats in R. The normalized vegetation (NDVI), normalized humidity (NDMI) and bare soil (BSI) indices had a correlation coefficient ≥ 0.8 in both landscapes, while evapotranspiration (EVA), solar radiation (SR) and elevation (ELV) were also significantly correlated in P2O (Table S3 in S2 File). Thus, we retained NDVI for the two landscapes and SR for P2O, to perform the connectivity analyses.

We estimated two genetic distances, $F_{ST}$ and the inverse of the proportion of shared alleles (*Dps*) between sampling sites within each landscape, using *pairwise.neifst* with hierfstat and *pairwise.propShared* with adegenet, respectively. $F_{ST}$ is a measure of population differentiation due to genetic structure, while *Dps* is more sensitive to recent demographic events [57,58]. Given that $F_{ST}$ is widely used in genetic studies, we include it as a metric that can be compared with other studies [59]. To evaluate isolation by distance (IBD) we performed two-dimensional IBD [60] with linear models, using linearized $F_{ST}$ and *Dps* and the logarithm of geographic (Euclidean) distance calculated with fossil 0.4.0 [61] in R.

We evaluated the influence of landscape features on genetic distances and established connectivity hypotheses (predictions to be tested; see Table S4 in S2 File), considering the biology and ecology of *R. horribilis*. Specifically, we selected

environmental variables that can potentially promote or impair individual movement. Considering that major roads and rivers are known to limit amphibian connectivity (e.g., [18]), we hypothesized that these features would function as barriers for *R. horribilis* movement and gene flow (Table S4 in S2 File). We assessed the relationship between genetic distance and both main roads and high-flow rivers that divide sampling sites within each landscape using two methods. We first performed a distance-based redundancy analysis (db-RDA) [62] that carries out a constrained ordination using non-Euclidian distance measures. We created a series of dummy variables (0, 1) to indicate that a sampling site was on one side of the road/river or the other, and used $F_{ST}$ and *Dps* values to perform principal coordinate analyses (PCoA). The resulting eingenvectors were then used as a response matrix in a RDA [63], with which we tested road/river 'barrier-effects'. We also applied a multiple regression on distance matrices (MRDM) incorporating multiple response variables [64] to test multiple barrier scenarios (see Fig 2). To do so, we built resistance surfaces for each barrier scenario, assigning a value of 1000 to roads and rivers and 1 to the rest of the landscape. We obtained least-cost distance matrices for each scenario (response variables) as estimated with *costDistance* in gdistance v.1.6.4 [65] in R, and the $F_{ST}$ and *Dps* matrices were used as exploratory variables. MRDM was run with *MRM* in ecodist v.2.0.9 [66] in R, with 10,000 permutations.

Although amphibian distribution is in general constrained by humidity [67], *R. horribilis* can be present across open areas and can hence tolerate harsh temperature and radiation conditions [31,32]. Likewise, *R. horribilis* is often found in areas with low vegetation cover [31], but rarely near/at urban cities, while bufonids in general prefer more humid habitats [68,69]. Accordingly, we hypothesized that extremely high temperature, evapotranspiration and solar radiation (i.e., detrimental conditions) will limit *R. horribilis* movement in areas with no vegetation cover (bare soil), while it will be facilitated in areas with higher vegetation cover (secondary forest and crops and pastures), humid soils and lower elevation; also, that impervious surfaces (urban areas) are expected to function as barriers to movement (Table S4 in S2 File). Lastly, given that water bodies are crucial for amphibian reproduction and can function as stepping-stones, facilitating the movement of individuals and connecting breeding sites [7,36,70], we predicted that areas with a higher probability for the presence of temporary water bodies (ponds and streams) will promote connectivity.

The connectivity models were performed with the optimization framework developed by Peterman [71], to determine the resistance values of our landscape variables surfaces with ResistanceGA [71,72]. This method uses a genetic algorithm (GA) [73] that adaptively explores the parameter space of monomolecular and Ricker functions [74] to transform continuous surfaces into resistance surfaces without prior assumptions. The monomolecular [$y = r\,(1\text{-exp}^{-bx})$] and Ricker [$y = r\,\text{exp}^{-bx}$] are two exponential-based functions used for ecological modeling, which differ in the curve shape of the relationship they are modeling. The curve shape is mainly determined by shape (x) and magnitude (b) parameters, which result in a saturating exponential (growth or decay) curve for the monomolecular function, and a hump-shaped curve (skewed to right or left) for the Ricker function [74]. During the optimization process, the genetic algorithm searches combinations of these parameters for transforming resistance surfaces, denoted by "r" in the monomolecular and Ricker equations [71,72]. The method also uses maximum-likelihood population effects mixed models (MLPE) to evaluate the fitness of each surface, considering the non-independence inherent to pairwise distance matrices. In the process, the GA seeks to maximize the relationship between pairwise genetic distances ($F_{ST}$ and *Dps*; response variable) and pairwise landscape distances (predictor variables), including the Euclidean distance model. First, surfaces were individually optimized using the *commuteDistance* function with gdistance in R, with three independent runs to verify convergence of parameters. The model fit statistic for the optimized resistance surfaces was the AICc (Akaike's information criterion corrected for small/finite sample size) [75], and a bootstrap resampling of the data to evaluate the robustness of the models. To do so, we randomly selected 75% of the samples without replacement and fitted each surface to each sample subset; the average rank, average $R^2$ (proportion of the variance explained by the fixed factors), and proportion (percentage) in which a surface was chosen as the best model were estimated with *Resist.boot* function [76], with 10,000 iterations. Finally, considering the best-supported individual models (see Table 1) and our connectivity hypotheses, we built four composite surfaces (aquatic, structural, biological-1 (facilitating) and biological-2 (limiting) models; see Table S5 in S2 File) to run

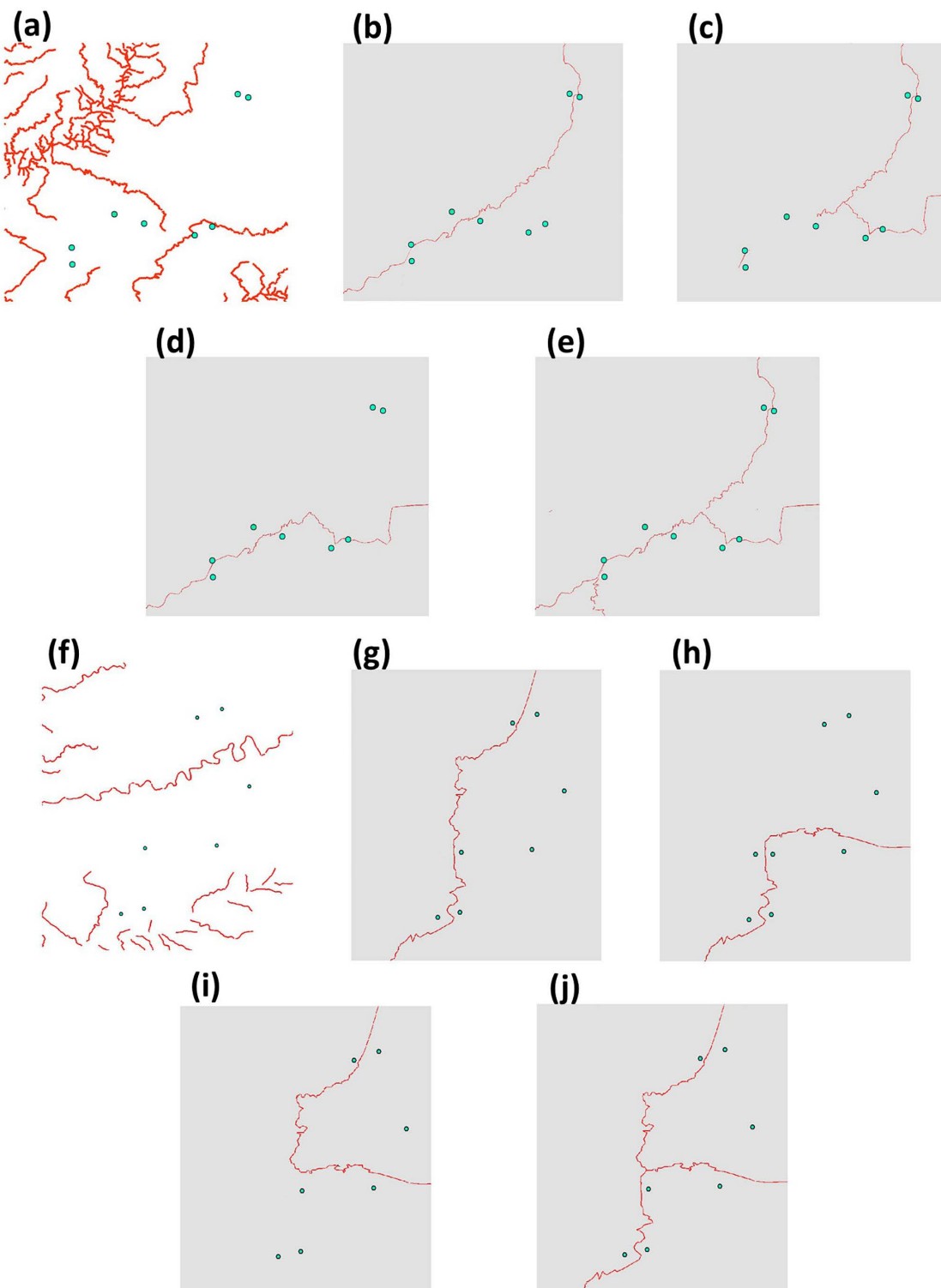

**Fig. 2. Main roads and rivers surfaces used in the multiple regression distance matrix (MRDM) to assess barrier isolation models by landscape. (a-e)** models for landscape 1 (P1O): **(a)** rivers; **(b)** road 1; **(c)** road 2; **(d)** road 3; **(e)** road 4. **(f-j)** models for landscape 2 (P2O): **(f)** rivers; **(g)** road 1; **(h)** road 2; **(i)** road 3; **(j)** road 4. The proposed barrier for each model is depicted in red.

**Table 1. Individual generalized linear mixed models based on $F_{ST}$ and *Dps* genetic distances for *Rhinella horribilis* in two landscapes (P1O and P2O) in Oaxaca, southern Mexico. Best-supported models are indicated by the highest 'top model' (%; refers to the percent of pseudo-bootstrap replicates where this model had the best fit). k: number of parameters fit in each model plus the intercept, AIC: Akaike information criterion. The average rank and average *R*2 (value of the fitted model; %) are shown.**

| Surface Landscape 1 (P1O) | k | AIC | ΔAIC | Average rank | Average $R^2$ (%) | Top model (%) |
|---|---|---|---|---|---|---|
| **Genetic distance ($F_{ST}$)** | | | | | | |
| Temporary streams (TS) | 4 | −104.24 | 0 | 3.03 | 79.82 | 39.39 |
| Solar radiation (SR) | 4 | −101.41 | −2.83 | 5.38 | 77.78 | 18.39 |
| Distance (D) | 2 | −99.42 | −4.82 | 6.67 | 67.39 | 0 |
| Temporary water bodies (TWB) | 4 | −99.85 | −4.40 | 7.17 | 75.77 | 14.43 |
| Land use (LU) | 7 | −100.33 | −3.91 | 7.39 | 84.05 | 20.83 |
| Ambient temperature (AT) | 4 | −98.79 | −5.45 | 7.96 | 72.37 | 0 |
| Elevation (ELV) | 4 | −98.27 | −5.97 | 8.96 | 67.71 | 3.31 |
| Evapotranspiration (EVA) | 4 | −98.05 | −6.20 | 9.35 | 70.01 | 0 |
| Relative humidity (RH) | 4 | −97.23 | −7.01 | 11.03 | 71.98 | 3.37 |
| Normalized vegetation (NDVI) | 4 | −96.94 | −7.31 | 11.22 | 69.21 | 0 |
| **Genetic distance (*Dps*)** | | | | | | |
| Ambient temperature (AT) | 4 | −132.13 | 0.00 | 4.41 | 65.73 | 27.67 |
| Distance (D) | 2 | −131.80 | −0.32 | 4.73 | 55.88 | 34.48 |
| Temporary streams (TS) | 4 | −130.87 | −1.25 | 6.77 | 62.6 | 12.43 |
| Solar radiation (SR) | 4 | −130.83 | −1.30 | 6.77 | 64.53 | 13.11 |
| Land use (LU) | 7 | −129.09 | −3.03 | 8.75 | 64.44 | 5.83 |
| Evapotranspiration (EVA) | 4 | −129.17 | −2.96 | 9.79 | 58.19 | 3.23 |
| Normalized vegetation (NDVI) | 4 | −129.26 | −2.87 | 9.85 | 57.94 | 3.26 |
| Relative humidity (RH) | 4 | −128.74 | −3.39 | 10.73 | 60.58 | 0 |
| Temporary water bodies (TWB) | 4 | −128.40 | −3.73 | 11.71 | 58.09 | 0 |
| Elevation (ELV) | 4 | −127.84 | −4.29 | 12.28 | 55.98 | 0 |
| **Surface Landscape 2 (P2O)** | k | AIC | ΔAIC | Average rank | Average R2 (%) | Top model (%) |
| **Genetic distance ($F_{ST}$)** | | | | | | |
| Normalized vegetation (NDVI) | 4 | −69.338 | 0.00 | 2.49 | 80.24 | 56.87 |
| Relative humidity (RH) | 4 | −66.949 | −2.39 | 3.21 | 75.33 | 9.96 |
| Temporary water bodies (TWB) | 4 | −63.931 | −5.41 | 4.87 | 41.94 | 18.61 |
| Distance (D) | 2 | −63.732 | −5.61 | 5.55 | 35.05 | 0 |
| Solar radiation (SR) | 4 | −63.013 | −6.33 | 5.93 | 53.35 | 5.04 |
| Ambient temperature (AT) | 4 | −62.402 | −6.94 | 6.42 | 47.89 | 9.53 |
| Temporary streams (TS) | 4 | −61.542 | −7.80 | 6.85 | 61.15 | 0 |
| Land use (LU) | 7 | −58.243 | −11.095 | 9.20 | 45.91 | 0 |
| **Genetic distance (*Dps*)** | | | | | | |
| Distance (D) | 2 | −76.97 | 0 | 2.08 | 43.43 | 85.86 |
| Relative humidity (RH) | 4 | −73.87 | −3.09 | 4.06 | 52.7 | 8.27 |
| Solar radiation (SR) | 4 | −73.76 | −3.20 | 4.66 | 51.49 | 1.85 |
| Ambient temperature (AT) | 4 | −73.29 | −3.68 | 5.26 | 49.33 | 4.02 |
| Normalized vegetation (NDVI) | 4 | −72.58 | −4.39 | 5.85 | 46.99 | 0 |
| Temporary streams (TS) | 4 | −72.92 | −4.05 | 5.93 | 45.35 | 0 |
| Temporary water bodies (TWB) | 4 | −73.03 | −3.94 | 6.21 | 44.37 | 0 |
| Land use (LU) | 7 | −67.49 | −9.48 | 9.65 | 48.61 | 0 |

a multivariate optimization approach. Optimization of composite surface models was conducted twice to ensure convergence [71]. Bootstrap model selection was performed again with the same previous parameters to obtain the average rank, average $R^2$, and the selection percentage of both univariate and multivariate surfaces.

## Results

A total of 190 *R. horribilis* adults were sampled, 125 in P1O and 65 in P2O (Table S2 in S2 File) and all were successfully sequenced. Genomic results yielded 1,134,057 polymorphic sites after filtering. The final filtered study-wide dataset included 6034 SNPs and the population-study-P1O and P2O datasets included 4088 and 4190 SNPs, respectively, both with an average coverage of 14x.

### Genetic structure and diversity

Genetic structure analyses with the study-wide dataset clearly identified two distinct genetic groups, as shown by the PCA and the SNMF structuring results, which clearly separate P1O and P2O (Fig 1d,e); additionally, results indicated a $F_{ST} = 0.0218$. Structuring results within landscapes (population level) identified three well defined genetic groups in P1O (Fig 3); all structure analyses (S2, S3A Fig in S1 File) separated sites in the north (ZAA and ZAB) in one group, and the

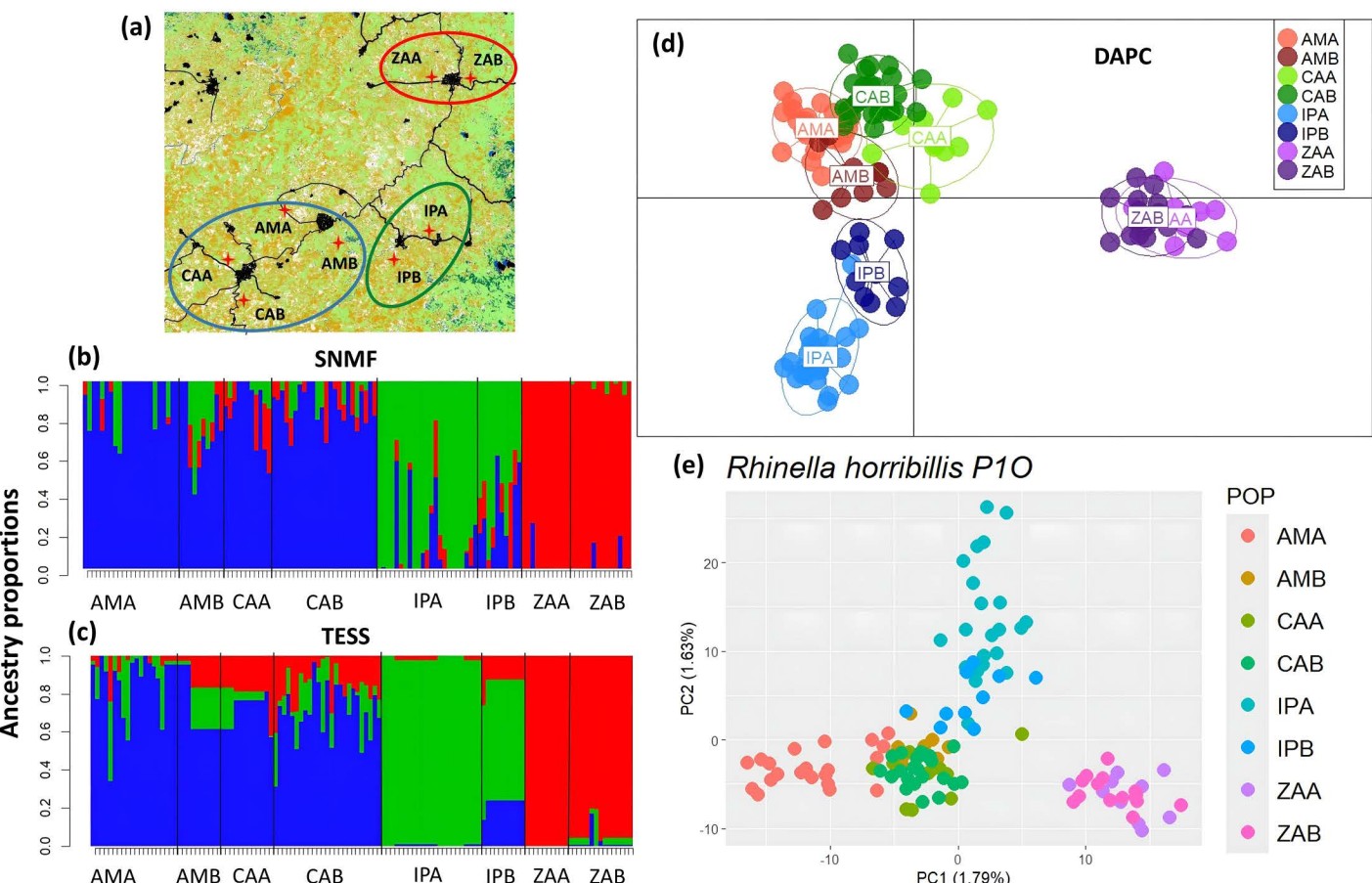

**Fig. 3. Genetic structure of *Rhinella horribilis* in landscape 1 (P1O) as estimated with different methods. (a)** Sampling sites in Oaxaca, southern Mexico, **(b)** SNMF (*K*=3), **(c)** TESS (*K*=3), **(d)** DAPC, **(e)** PCA.

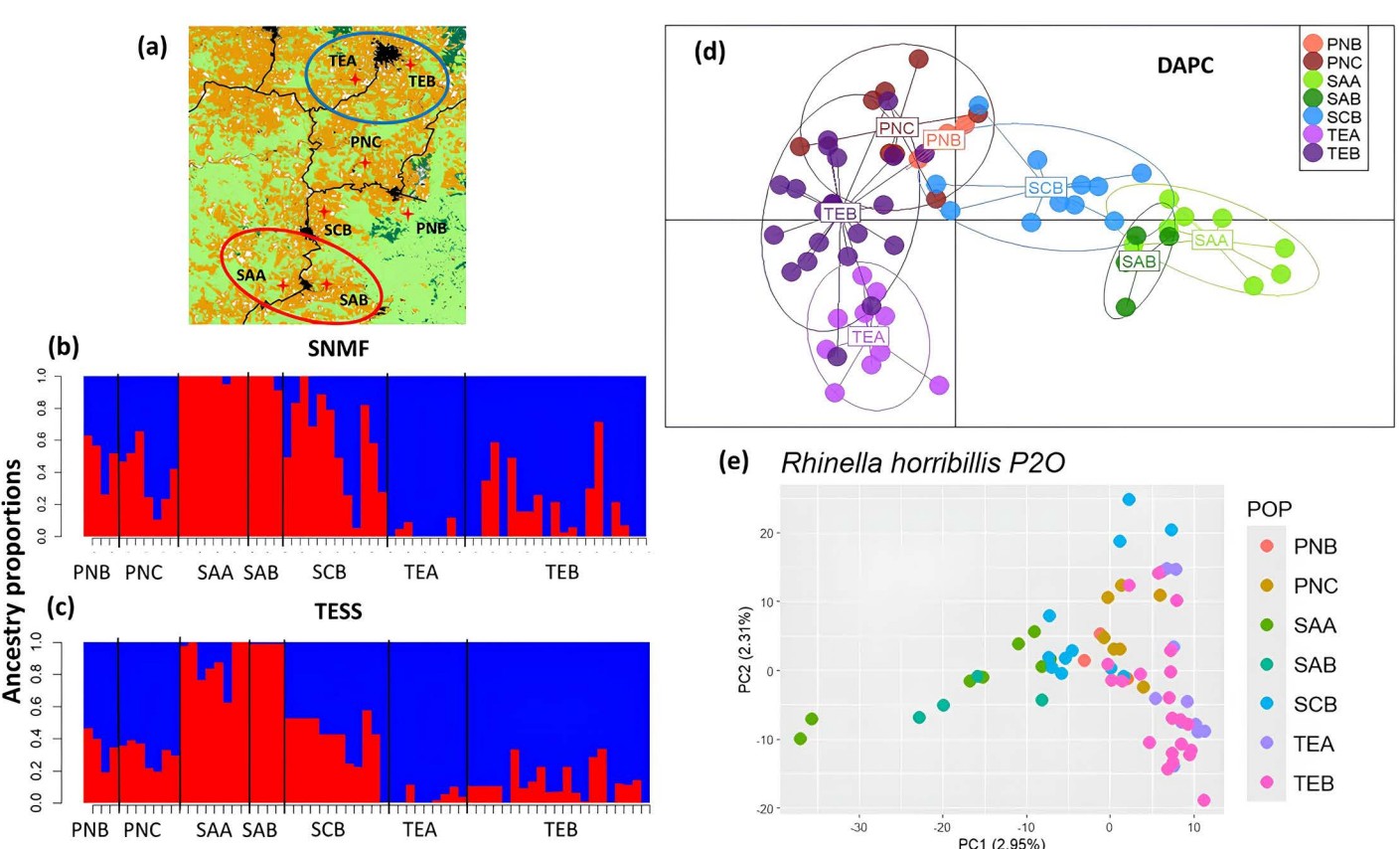

more central-southern ones in two groups (IPA and IPB) and (CAA, CAB, AMA, AMB), which exhibit little levels of admixture. SNMF and TESS also suggested *K* = 4, separating the sampling sites CAA-CAB and AMA-AMB (S2B, S2E Fig in S1 File). Importantly, we corroborated that each sampling site did not form a distinct genetic group, implying that the roads and rivers separating each pair of sites do not impede gene flow (S2C, S2F Fig in S1 File). Structuring results for P2O (*K* = 2–3) more consistently identified two genetic groups, one in the northern (TEA and TEB) and one in the southern (SAA and SAB) zones of the study region, with the three remaining sites admixed with both groups (Fig 4, S4 Fig in S1 File); only DAPC (Fig 4d) and sPCA (S3B Fig in S1 File) suggest these central sites as a potential genetic group. As for P1O, exploring higher values of *K* did not provide further genetic structure resolution (S4A-S4F Fig in S1 File).

Overall genetic differentiation ($F_{ST}$) within landscapes was higher in P1O than in P2O (mean $F_{ST}$ = 0.0246, P1O; $F_{ST}$ = 0.0218, P2O), while *Dps* showed higher values in P2O (mean *Dps* = 0.0672) compared to P2O (0.0546). Results based on $F_{ST}$ were concordant with the observed structure within landscapes. In P1O, the sites Santa María Zacatepec-B and A (ZAB, ZAA) and Santa Maria Ipalapa-A (IPA) were the most differentiated from the rest, while in P2O, the highest differentiation was observed between Santa Ana Tututepec-A and B (SAA, SAB) from the rest (Table S6 in S2 File). Differentiation patterns between sites were less clear with *Dps* in both landscapes (Table S6 in S2 File).

Finally, genetic diversity (average observed and expected heterozygosity) was significantly (*p* < 0.001) higher in P2O (*Ho* = 0.139 ± 0.0040, *He* = 0.153 ± 0.0049) than P1O (*Ho* = 0.109 ± 0.0022, *He* = 0.118 ± 0.0030), *t*-value = −18.211 and

**Fig 4. Genetic structure of *Rhinella horribilis* in landscape 2 (P2O) as estimated with different methods. (a)** Sampling sites in Oaxaca, southern Mexico, **(b)** SNMF (*K* = 2), **(c)** TESS (*K* = 2), **(d)** DAPC, **(e)** PCA.

*t-value*=−15.706, respectively. In all sampling sites, the observed heterozygosity was lower than the expected heterozygosity, while low $F_{IS}$ values were inferred except for IPA in P1O, and SCB, TEB and PNC in P2O (Table S2 in S2 File), with not statistical differences between landscapes (*t-value*=−1.753, *p*=0.103).

## Landscape genetics

Results indicated significant isolation by distance in both P1O ($F_{ST}$: $R^2$=0.559, *Dps*: $R^2$=0.527; *p*<0.001) and P2O ($F_{ST}$: $R^2$=0.373; *Dps*: $R^2$=0.495; *p*=0.003) landscapes (S5 Fig in S1 File). Regarding the testing of roads and rivers as barriers to gene flow, in P1O the db-RDA based both on $F_{ST}$ and *Dps* showed a positive relationship, supporting that a river that crosses in the north (River 2) effectively separates the north genetic group from the rest of localities (Table 2). Likewise, MRDM showed a positive relationship between $F_{ST}$ and the roads model 4 (the one that considers all the roads in the landscape; Fig 2e) (F=0.7962, *p*=0.037), suggesting that these roads limit connectivity. Comparatively, in P2O the MRDM with $F_{ST}$ showed a negative relationship in the roads model 4 (Fig 2j) (F=−1.4306, *p*=0.005), suggesting that these roads facilitate connectivity among sampling sites. No other variables in the two landscapes were identified as relevant in explaining genetic structure (*p*>0.05; Table 2).

Model selection after optimization of univariate models based on $F_{ST}$ genetic distance showed that the temporary streams (TS) was the best-supported model (39.39%; ΔAIC=0, $R^2$=79.82%) in P1O (Table 1); this model exhibited an Inverse monomolecular Ricker function (Fig 5a) and assigned low resistance (high connectivity) from low to intermediate probability of presence of TS (<0.5), with a fast increase in resistance at higher values (>0.5). Also with high support

**Table 2. Results of isolation by barrier (IBB) models for *Rhinella horribilis* in two landscapes (P1O and P2O) in Oaxaca, southern Mexico, using $F_{ST}$ and *Dps* for the distance-based RDA (db-RDA) and regression on distance matrices (MRDM). Significant results are highlighted in bold (*p*≤0.05). See Fig 2 for details of each tested model.**

| db-RDA | | | | MRDM | | | |
|---|---|---|---|---|---|---|---|
| **Landscape** | | | | **Landscape** | | | |
| | **Barrier** | **F** | **p** | | **Barrier** | **F** | **p** |
| **P1O $F_{ST}$** | Road 1 | 0.6516 | 0.656 | **P1O *FST*** | Road 1 | 0.0748 | 0.3677 |
| | Road 2 | 0.2422 | 0.946 | | Road 2 | 0.0506 | 0.5248 |
| | Road 3 | 1.3866 | 0.217 | | Road 3 | 0.0517 | 0.5048 |
| | River 1 | 1.5762 | 0.118 | | Road 4 | 0.7962 | **0.0367*** |
| | River 2 | 2.9604 | **0.047*** | | Rivers | 0.0828 | 0.5967 |
| **P2O $F_{ST}$** | Road 1 | 0.3878 | 0.758 | **P2O *FST*** | Road 1 | 0.5077 | 0.0644 |
| | Road 2 | 0.4342 | 0.706 | | Road 2 | 0.0422 | 0.8388 |
| | Road 3 | 0.3867 | 0.800 | | Road 3 | 0.4192 | 0.0704 |
| | River 1 | 1.6717 | 0.190 | | Road 4 | −1.4306 | **0.0046*** |
| | | | | | Rivers | 0.1760 | 0.5320 |
| **P1O *Dps*** | Road 1 | 0.9036 | 0.669 | **P1O *Dps*** | Road 1 | 0.0627 | 0.1671 |
| | Road 2 | 0.7705 | 0.952 | | Road 2 | 0.0237 | 0.5782 |
| | Road 3 | 1.0132 | 0.436 | | Road 3 | 0.0733 | 0.0885 |
| | River 1 | 1.2529 | 0.063 | | Road 4 | −0.0225 | 0.9207 |
| | River 2 | 1.3831 | **0.05*** | | Rivers | −0.0119 | 0.8831 |
| **P2O *Dps*** | Road 1 | 0.7982 | 0.801 | **P2O *Dps*** | Road 1 | 1277.4 | 0.2529 |
| | Road 2 | 0.9326 | 0.581 | | Road 2 | −0.0090 | 0.9475 |
| | Road 3 | 1.0587 | 0.493 | | Road 3 | −0.0157 | 0.9069 |
| | River 1 | 1.5217 | 0.091 | | Road 4 | −0.1466 | 0.5474 |
| | | | | | Rivers | 0.2664 | 0.1514 |

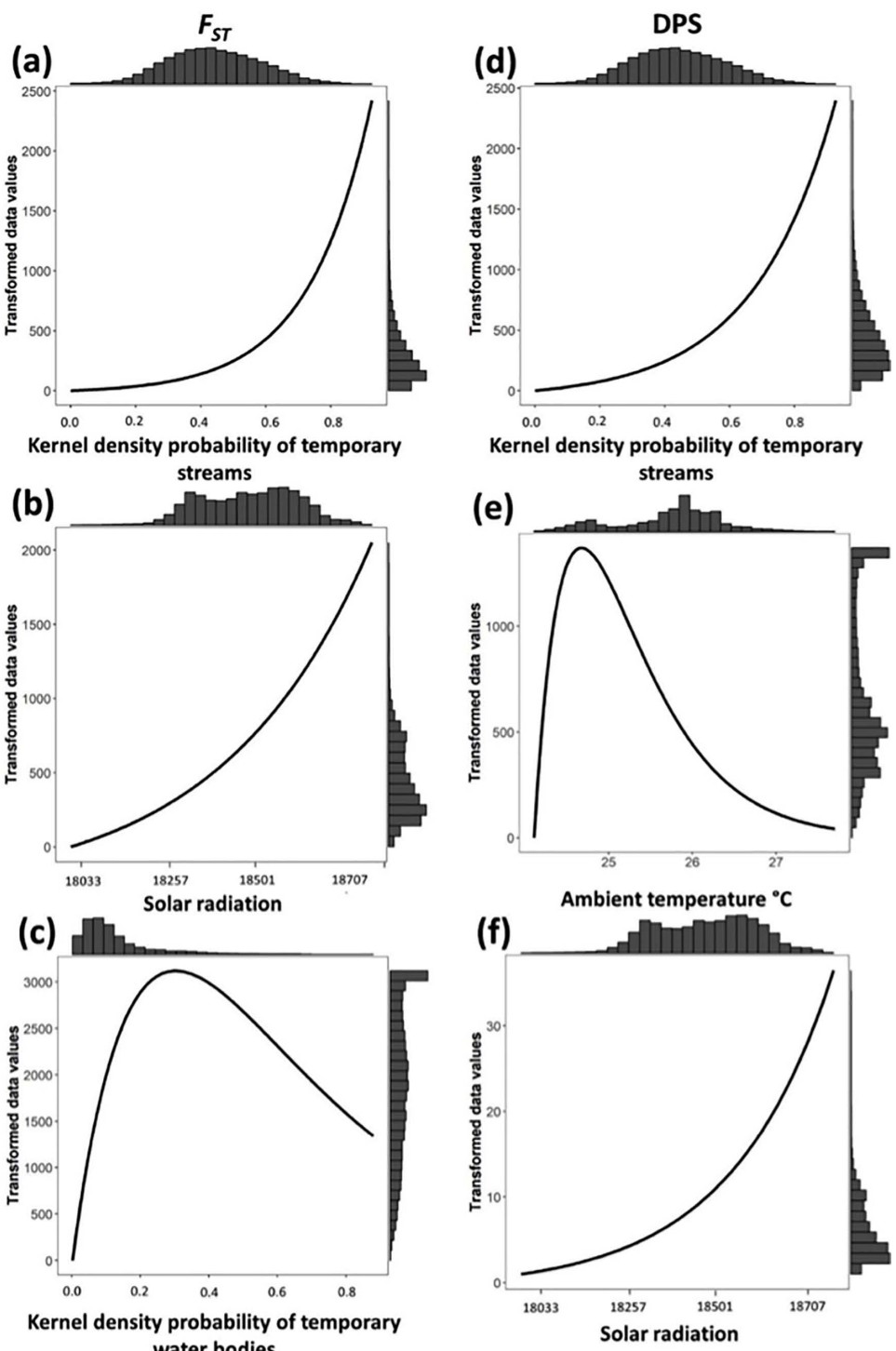

**Fig 5. Single surface optimization response curves for landscape 1 (P1O).** Plots **a-c** (left side) are based on $F_{ST}$ and plots **d-f** (right side) on *Dps* genetic distance. Each curve represents the resistance cost imposed by each landscape variable after the optimization procedure. Histograms represent the frequency of each resistance value. (**a**) and (**d**) temporary streams (TS), (**b**) and (**f**) solar radiation (SR), (**c**) temporary water bodies (TWB) and (**e**) ambient temperature (AT).

were solar radiation (SR, 18.39%), assigning low resistance below 18200 kJ m-2 day-1, and temporary water bodies (TWB, 14.43%) that assigned low resistance values at low and high probability of presence of TWB (Fig 5b,c). Land use showed high support (LU, 20.83%) and urban area, crops and pastures, secondary forest, and temporary water bodies had the lowest resistance values (Table S7 in S2 File). In P2O, the normalized vegetation index (NDVI; 56.87%, ΔAIC = 0, $R^2$ = 80.24%) and temporary water bodies (TWB; 18.61%, ΔAIC = −5.41, $R^2$ = 41.94%) were the best-supported models, followed by relative humidity (RH, 9.96%; ΔAIC = −2.39, $R^2$ = 75.33%) (Table 1), exhibiting Inverse Ricker functional forms. The NDVI model assigned low resistance to low to mid vegetation at values between 0.2 and 0.6 (Fig 6a); the probability of formation of TWB < 0.6 also had low resistance (Fig 6b), while RH assigned high resistance to high values (>88%) (Fig 6c).

Univariate models based on *Dps* showed that the best-supported model in P1O was geographic distance (D; 34.48%, ΔAIC = −0.32, $R^2$ = 55.88%); however, other variables were equally likely (ΔAIC <4) and explained higher variation ($R^2$) than D (Table 1). Namely, ambient temperature (AT; 27.67%, ΔAIC = 0, $R^2$ = 65.73%) that exhibited a Ricker functional form and assigned lower resistance to high values (>26°C); TS (12.45%, ΔAIC = −1.25, $R^2$ = 62.69%) and SR (15.11%, ΔAIC = −1.30, $R^2$ = 64.53%) that exhibited the same response curves and assigned, respectively, low resistance at low probability of presence of TS (<0.4) and below 18300 kJ m-2 day-1 SR (Fig 5d-f). LU showed lower support (5.83%), with similar resistance values as that for $F_{ST}$ genetic distance (Table S7 in S2 File). Similar results were found for P2O (Table 1), where distance (D; 85.8%, ΔAIC = 0, $R^2$ = 43.4%) was the best-supported model and equally likely variables (ΔAIC <4) with higher $R^2$ were RH (8.3%, ΔAIC = −3.09, $R^2$ = 52.7%), AT (4.02%, ΔAIC = −3.68, $R^2$ = 49.3%) and SR (1.85%, ΔAIC = −3.2, $R^2$ = 51.5%), exhibiting each low resistance values below 84%, above 25°C and above 18500 kJ m-2 day-1 (Fig 6d-f).

The multivariate model selection showed for both landscapes (P1O and P2O) and both genetic distances ($F_{ST}$ and *Dps*) that the Aquatic model (Temporary water bodies + Temporary streams) was the best-supported model and the one with highest genetic variation explained (Table 3). Additionally, P2O exhibited the Structural model (Temporary water bodies + Temporary streams + NDVI) equally likely to explain $F_{ST}$ variation (ΔAICc < 4). Accordingly, the resistance surface of the Aquatic models for P1O and both genetic distances (S6A, S6B Fig in S1 File) indicated high resistance (limited connectivity) between the northernmost sampling sites (ZAA-ZAB) and the rest (red circle); and to a lesser extent between the central IPA-IPB (green circle) sites. The Aquatic models for P2O showed inverse patterns for each genetic distance, indicating high resistance between northern and southern genetic groups (TEA-TEB, red circle and SAA-SAB, purple circle; S6C Fig in S1 File) for $F_{ST}$; the latter also supported by the high resistance identified by the structural model (S6D Fig in S1 File). While for *Dps*, the Aquatic model indicated low resistance among all localities (S6E Fig in S1 File).

## Discussion

Unlike its close relative, the amply known Cane Toad *Rhinella marina,* the Giant Toad *R. horribilis* has been little studied, particularly concerning population genetic assessments within its native distribution. Our findings show that the landscapes studied are genetically differentiated given their distinct degree of habitat modification. Furthermore, at the local level (within landscapes), our results demonstrate that functional connectivity is significantly influenced by barriers like rivers and roads, and that water bodies availability was the most important landscape feature for *R. horribilis* connectivity, while vegetation cover, solar radiation and relative humidity also played an important role.

### Interplay of landscape features on genomic patterns and functional connectivity

As predicted, the two landscapes (P1O and P2O) studied are genetically differentiated, supporting the fact that *R. horribilis* populations are not connected. Low levels of genetic variability and increased inbreeding are often observed in wild animal populations inhabiting fragmented and anthropogenically modified habitats [10,77], and amphibians are no exception [19,78–80]. However, compared to those studies [19,78–80], we observed overall similar genetic diversity in the *R.*

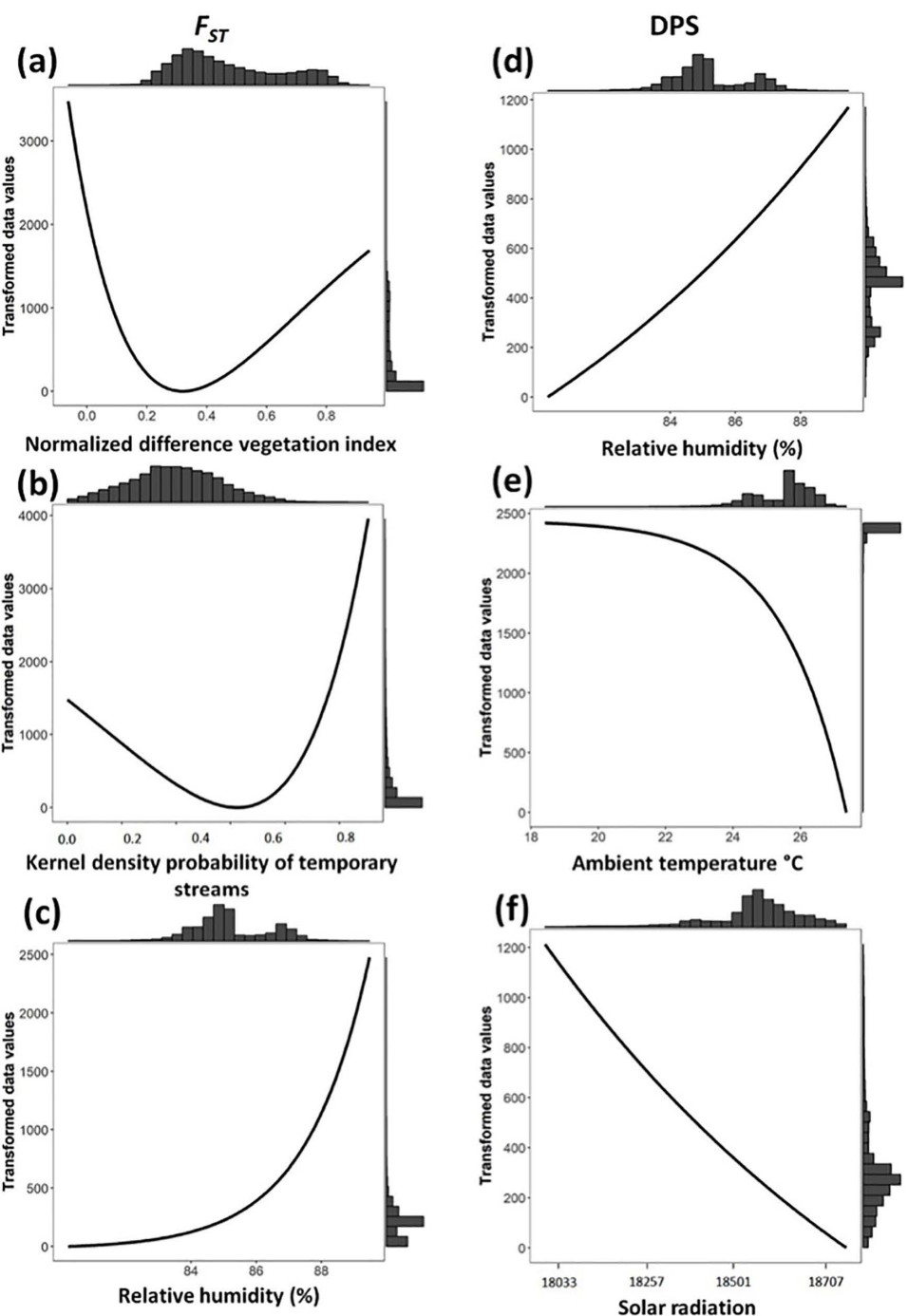

**Fig 6. Single surface optimization response curves for landscape 2 (P2O).** Plots **a-c** (left side) are based on $F_{ST}$ and plots **d-f** (right side) on *Dps* genetic distance. Each curve represents the resistance cost imposed by each landscape variable after the optimization procedure. Histograms represent the frequency of each resistance value. (**a**) Normalized differences vegetation index (NDVI), (**b**) temporal water bodies (TWB), (**c**) and (**d**) relative humidity (RH), (**e**) ambient temperature (AT), and (**f**) solar radiation (SR).

**Table 3. Multivariate generalized linear mixed models based on $F_{ST}$ and *Dps* genetic distance for *Rhinella horribilis* in two landscapes (P1O and P2O) in Oaxaca, southern Mexico.** Best-supported models are indicated by highest 'top model' (%; refers to the percent of pseudo-bootstrap replicates where this model had the best fit) in bold; AICc: Akaike information criterion. The average rank and average R2 (value of the fitted model; %) are shown.

| Model | Surface | AICc | ΔAICc | Average rank | Average $R^2$ | Top model (%) |
|---|---|---|---|---|---|---|
| **P1O** | **Genetic distance $F_{ST}$** | | | | | |
| **Aquatic model** | **Temporary water bodies** | **−98.13** | **0** | 1.07 | **79.81** | **92.98** |
| | **Temporary streams** | | | | | |
| Structural | Temporary water bodies | −92.10 | −6.03 | 2.43 | 79.8 | 0 |
| | Temporary streams NDVI | | | | | |
| Biological model 2 | BSI | −89.50 | −8.63 | 2.60 | 77.48 | 7.02 |
| | Temperature | | | | | |
| | Solar radiation | | | | | |
| Biological model 1 | Temporary water bodies Temporary streams | −82.00 | −16.13 | 3.89 | 75.79 | 0 |
| | NDMI | | | | | |
| | RH | | | | | |
| **P1O** | **Genetic distance *Dps*** | | | | | |
| **Aquatic model** | **Temporary water bodies** | **−125.15** | **0** | **1.10** | **63.72** | **89.52** |
| | **Temporary streams** | | | | | |
| Biological model 2 | BSI | −120.30 | −4.854 | 1.97 | 67.27 | 10.48 |
| | Temperature | | | | | |
| | Solar radiation | | | | | |
| Structural | Temporary water bodies Temporary streams | −117.07 | −8.084 | 2.93 | 59.6 | 0 |
| | NDVI | | | | | |
| Biological model 1 | Temporary water bodies | −111.26 | −13.893 | 4 | 62.03 | 0 |
| | Temporary streams | | | | | |
| | NDMI | | | | | |
| | RH | | | | | |
| **P2O** | **Genetic distance $F_{ST}$** | | | | | |
| **Structural** | **Temporary water bodies** | **−45.61** | **0** | **1.81** | **64.46** | **48.15** |
| | **Temporary streams** | | | | | |
| | **NDVI** | | | | | |
| **Aquatic model** | **Temporary water bodies** | **−43.53** | **−2.08** | **1.87** | **27.50** | **51.85** |
| | **Temporary streams** | | | | | |
| Biological model 2 | BSI | −40.09 | −5.52 | 2.99 | 42.32 | 0 |
| | Temperature | | | | | |
| | Solar radiation | | | | | |
| Biological model 1 | Temporary water bodies | −39.50 | −6.11 | 3.33 | 77.46 | 0 |
| | Temporary streams | | | | | |
| | NDMI | | | | | |
| | RH | | | | | |
| **P2O** | **Genetic distance *Dps*** | | | | | |
| **Aquatic model** | **Temporary water bodies** | **−66.12** | **0** | **1.10** | **45.02** | **90.46** |
| | **Temporary streams** | | | | | |
| Biological model 2 | BSI | −61.70 | −4.42 | 2.24 | 51.21 | 9.54 |
| | Temperature | | | | | |
| | Solar radiation | | | | | |

*(Continued)*

**Table 3.** (Continued)

| Model | Surface | AICc | ΔAICc | Average rank | Average $R^2$ | Top model (%) |
|-------|---------|------|-------|--------------|---------------|---------------|
| Structural | Temporary water bodies | −60.53 | −5.59 | 2.71 | 48.44 | 0 |
| | Temporary streams | | | | | |
| | NDVI | | | | | |
| Biological model 1 | Temporary water bodies | −56.49 | −9.63 | 3.96 | 56.86 | 0 |
| | Temporary streams | | | | | |
| | NDMI | | | | | |
| | RH | | | | | |

*horribilis* populations studied. Similar to our findings, Arruda and collaborators [81] report moderate genetic diversity and low inbreeding (measured with microsatellite loci) in populations of the Cururu Toad *Rhinella schneideri* inhabiting agricultural areas. Such patterns may be related with certain life history traits of the two species, including large body size and high dispersal ability [82], which allow them to occupy a wide variety of habitats [32].

Notably, we found as expected lower genetic diversity, higher genetic structure and lower functional connectivity among populations in the landscape with higher habitat modification (P1O), also exhibiting isolation by barrier associated with the main river and roads (the latter in P2O as well). Amphibian species with low vagility usually exhibit more structured populations in modified compared to natural, well-preserved areas [16,17,19,80]. In contrast, species with greater mobility may counter these effects, for example cases like *R. schneideri* and the Southern Leopard Frog *Rana [Lithobates] sphenocephalus*, which show low genetic differentiation in agriculture and urban landscapes, respectively [81,83]. Interestingly, although *Rhinella* species are characterized by high dispersal capacity (up to 2 km home ranges) [84], our results show some degree of fine-scale genetic structure in *R. horribilis,* suggesting there are other factors in place influencing individual dispersal patterns. For instance, individual dispersal at these fine scales can be associated with sites that provide favorable conditions for shelter, feeding and reproduction [31,85,86], like the temporary water bodies in our study sites distributed in pasture lands near patches of high cover vegetation.

Indeed, water bodies were the most important drivers of functional connectivity for *R. horribilis* in both landscapes. The second order river (i.e., large, strong current) that divides the northern (Santa María Zacatepec) from the rest of sampling sites in P1O limited dispersal and gene flow, influencing the observed genetic structure. These results are consistent with previous studies in other amphibian species across a wide range of landscape types, showing that the magnitude, direction and intensity of water flow can have a strong impact on patterns of gene flow at the landscape scale [18,87–90].

Temporary water bodies (streams and ponds) were crucial for maintaining functional connectivity associated with the modified environment in both landscapes. Our findings showed an overall gradient where resistance was lower when water bodies were not very abundant (<0.5 in a scale from 0 to 1), with resistance rapidly increasing at higher values (>0.5). Low to medium density of temporary water bodies (natural and artificial), which are frequent in modified landscapes, enable individual stepping-stone dispersal, facilitating genetic interchange and likely sustaining the genetic diversity we observe in *R. horribilis* populations. These results agree with previous studies showing that temporary streams associated with riparian vegetation and forested areas promote connectivity in urban and agricultural areas (e.g., [19,20,86,87,91–93]). Our findings highlight the importance of temporary water bodies for dispersal and reproduction in *R. horribilis* in both landscapes, irrespective of the degree of habitat modification. Therefore, it is crucial to conserve these landscape features to maintain populations of this and other anuran species. Building artificial water bodies surrounded by vegetation cover in sites with extensive agriculture and livestock areas can be fundamental to mitigate the negative impacts of land use and climate changes on amphibian populations.

Vegetation cover, solar radiation and relative humidity were key landscape features influencing *R. horribilis* connectivity. This was supported by our model selection results yet, interestingly, with different patterns in each landscape associated with their degree of modification. Landscape P2O is characterized by more vegetation cover, lower fragmentation and less agriculture, pastures and livestock land use areas (Fig 1). Accordingly, *R. horribilis* in P2O exhibited higher connectivity associated with low to moderate vegetation cover, as well as with high relative humidity (up to 88%) and ambient temperature (25−27ºC), while high resistance was observed with high solar radiation values (above 18500 kJ m-2 day-1). Similar connectivity patterns have been documented for amphibians that occupy sites with moderate vegetation cover, including the semiaquatic salamanders *Hynobius yangi*, the terrestrial frogs *Ascaphus montanus* and *A. truei*, and tree frogs [19,88,93,94]. In contrast, lower solar radiation (below 18300 kJ m-2 day-1), higher AT (>26ºC) and less vegetation cover (grassland/shrubs to bare soil) were significantly associated with connectivity in landscape P1O. Large terrestrial anurans like *R. horribilis* can use behavioral and physiological strategies to disperse through open areas and avoid water loss, using temporary water bodies and areas with high humidity.

### Future directions

Given that temporary water bodies were a relevant feature for *R. horribilis* connectivity, it would be interesting to test specific hypotheses related to variables of the hydrological network and its temporal variability, to evaluate their effect on functional connectivity of multiple anuran species. Also, prioritizing the management of water bodies would be essential to sustain amphibian population dynamics, enhancing individual movement and genetic exchange.

Anthropized environments can exert multiple selective pressures on wild amphibian populations, therefore identifying cases of local adaptation can illuminate how populations deal with those stressful conditions. Hence, comparative studies on syntopic species should be conducted to assess if they respond similarly at the genomic level to analogous selection pressures, and to characterize key landscape features for functional connectivity and survival. It would be interesting to expand our study with *R. horribilis*, to determine if it presents signals of parallel evolution in landscapes representing different degrees of human modification. Also, comparing such gradient with populations from areas with no anthropogenic impacts to assess if their populations exhibit genetic variants that are fixed or specifically associated to natural (conserved) landscapes. Identifying genes and metabolic pathways and their associated functions would be crucial for amphibian adaptation to anthropized environments, which likely allow individuals to tolerate adverse environmental and water conditions. In this regard, it would be interesting to evaluate the expression of upregulated genes associated with such pathways and the molecular mechanisms acting on tadpole development and survival in mesocosm experiments or common garden experiments.

### Supporting information

**S1 File. Supplementary figures S1-S6.**
(PDF)

**S2 File. Supplementary tables S1-S7.**
(PDF)

**S3 File. Supplementary Methods.**
(PDF)

### Acknowledgments

We are grateful to R. Palacios, H. Colín, S. Hernández, R. Peralta and R. López for their enthusiastic help during fieldwork, and to N. Gálvez-Reyes for providing molecular laboratory assistance. We deeply thank the people from the field sites in Oaxaca for allowing us to work on their ejidos and private properties. Gerardo J. Soria-Ortiz acknowledges that

this paper was a part of his doctoral thesis in the Programa de Doctorado de Ciencias Biológicas de la Universidad Nacional Autónoma de México (UNAM). He was granted a scholarship from

Consejo Nacional de Ciencia y Tecnología (CONACyT CVU: 814369/No. Beca: 464869), which has now become the Secretaría de Ciencia, Humanidades, Tecnología e Innovación (SECIHTI), and support from Programa de Estudios de Posgrado (PAEP) and UNAM. Open access was obtained thanks to the UNAM agreement that covers article publication charges.

## Author contributions

**Conceptualization:** Gerardo J. Soria-Ortiz, Juan P. Jaramillo-Correa, Ella Vázquez-Domínguez.

**Formal analysis:** Gerardo J. Soria-Ortiz.

**Funding acquisition:** Leticia M. Ochoa-Ochoa, Ella Vázquez-Domínguez.

**Investigation:** Gerardo J. Soria-Ortiz, Leticia M. Ochoa-Ochoa, Ella Vázquez-Domínguez.

**Methodology:** Gerardo J. Soria-Ortiz, Leticia M. Ochoa-Ochoa, Juan P. Jaramillo-Correa, Íñigo Martínez-Solano, Ella Vázquez-Domínguez.

**Supervision:** Leticia M. Ochoa-Ochoa, Juan P. Jaramillo-Correa, Ella Vázquez-Domínguez.

**Writing – original draft:** Gerardo J. Soria-Ortiz, Juan P. Jaramillo-Correa, Íñigo Martínez-Solano, Ella Vázquez-Domínguez.

**Writing – review & editing:** Gerardo J. Soria-Ortiz, Leticia M. Ochoa-Ochoa, Juan P. Jaramillo-Correa, Íñigo Martínez-Solano, Ella Vázquez-Domínguez.

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
