## [Decision Letter · Decision Letter 0]

31 Mar 2025

PONE-D-25-03029Concordant yet unique neutral and adaptive genomic responses to anthropogenically modified landscapes in Rhinella horribilisPLOS ONE

Dear Dr. Vázquez-Domínguez,

Thank you for submitting your manuscript to PLOS ONE. After careful consideration, we feel that it has merit but does not fully meet PLOS ONE’s publication criteria as it currently stands. Therefore, we invite you to submit a revised version of the manuscript that addresses the points raised during the review process.

The study system is quite interesting, and the RADseq approach has power to detect population structure and possibly also signs of positive selection. The reviewers had mainly issues with analyses, from SNP filtering up to genotype to enviroment analyses. We acknowledge that it is interesting to aim to screen for loci under selection in two sets of populations, though there is some doubt about how best to do this and if it is achievable with this data. The main critiques by the reviewers are listed below, some of them demand substantial work. We ask you to navigate their comments and take a decision on how best to restructure the paper after the adjusted filtering and new analyses. Note, you dont have to do all they request, a scaled back version dropping the last section (genotype to environment association tests or tests of parallel selection) would still have merit.

The main concerns are.

1.    How comparable are the RADseq datasets? Recommend reanalyses of population structure of the entire set, see reviewer 2 comments. (if there are issues with the overlap, restrict the analyses to several hundred – thousand best overlapping markers).

2.    Better describe the filtering (did you use genotype depth, genotype quality, or allele balance filters?) and consider running analyses with and without the stringest HWE filter (rev 1)

3.    There are concerns about the power of the FST approach (reviwers recommend alterntive analytical approaches) and if the relatively low SNP coverage in the two datasets makes it unrealistic to actually detect loci under positive selection.

4.    The genotype to environment associations could be influenced (masked or enhanced) by population structure.

5.    The raw data have to be released on SRA or equivalent.

6.    Scripts detailing analyses should also be available, Github or similar.

We look forward to receiving your revised manuscript.

Kind regards,

Arnar Palsson, Ph.D.

Academic Editor

PLOS ONE

Journal Requirements:

LMOO and EVD acknowledge project funding from Consejo Nacional de Humanidades, Ciencias y Tecnologías CONAHCyT (grant # PN 2271). LMOO also acknowledges project funding from Programa de Apoyo a Proyectos de Investigación e Innovación Tecnológica (PAPIIT-DGAPA IN220321).  

4. We note that Figures 1, 2 and 3 in your submission contain [map/satellite] images which may be copyrighted. All PLOS content is published under the Creative Commons Attribution License (CC BY 4.0), which means that the manuscript, images, and Supporting Information files will be freely available online, and any third party is permitted to access, download, copy, distribute, and use these materials in any way, even commercially, with proper attribution. For these reasons, we cannot publish previously copyrighted maps or satellite images created using proprietary data, such as Google software (Google Maps, Street View, and Earth). For more information, see our copyright guidelines: http://journals.plos.org/plosone/s/licenses-and-copyright.

a. You may seek permission from the original copyright holder of Figures 1, 2 and 3 to publish the content specifically under the CC BY 4.0 license.  

Additional Editor Comments:

The study system is quite interesting, and the RADseq approach has power to detect population structure and possibly also signs of positive selection. The reviewers had mainly issues with analyses, from SNP filtering up to genotype to environment analyses. We acknowledge that it is interesting to aim to screen for loci under selection in two sets of populations, though there is some doubt about how best to do this and if it is achievable with this data. The main critiques by the reviewers are listed below, some of them demand substantial work. We ask you to navigate their comments and take a decision on how best to restructure the paper after the adjusted filtering and new analyses. Note, you dont have to do all they request, a scaled back version dropping the last section (genotype to environment association tests or tests of parallel selection) would still have merit.

The main concerns are.

1. How comparable are the RADseq datasets? Recommend reanalyses of population structure of the entire set, see reviewer 2 comments. (if there are issues with the overlap, restrict the analyses to several hundred – thousand best overlapping markers).

2. Better describe the filtering (did you use genotype depth, genotype quality, or allele balance filters?) and consider running analyses with and without the stringest HWE filter (rev 1)

3. There are concerns about the power of the FST approach (reviwers recommend alterntive analytical approaches) and if the relatively low SNP coverage in the two datasets makes it unrealistic to actually detect loci under positive selection.

4. The genotype to environment associations could be influenced (masked or enhanced) by population structure.

5. The raw data have to be released on SRA or equivalent.

6. Scripts detailing analyses should also be available, Github or similar.

Reviewers' comments:

Reviewer's Responses to Questions

**Comments to the Author**

1. Is the manuscript technically sound, and do the data support the conclusions?

Reviewer #1: Partly

Reviewer #2: Partly

2. Has the statistical analysis been performed appropriately and rigorously? 

Reviewer #1: No

Reviewer #2: Yes

3. Have the authors made all data underlying the findings in their manuscript fully available?

Reviewer #1: No

Reviewer #2: No

4. Is the manuscript presented in an intelligible fashion and written in standard English?

Reviewer #1: Yes

Reviewer #2: Yes

5. Review Comments to the Author

Reviewer #1: My feedback exceeds 20,000 characters, so I have attached it as a PDF. Accordingly, the remaining space here is being used to meet the minimum character limit of two hundred characters. Please refer to the attached PDF for detailed feedback.

Reviewer #2: Soria-Ortiz et al. used RADseq in a large sample (n = 190) of Rhinella horribilis from two geographic regions in Oaxaca, Mexico, to generate the first population genomic characterization of this species. The two geographic regions that the authors focused on differed substantially in their degree of anthropogenic land transformation, which enabled the authors to make inferences about how human-driven environmental change affects genome-wide diversity and genetic connectivity. Furthermore, the authors used multiple approaches to identify functional loci likely involved in adaptation to the different environments, highlighting the NOTCH and MAPK pathways as likely being key to adaptation to anthropogenetically modified habitats in R. horribilis, and possibly other amphibians. I think that this is extremely valuable and insightful work from both an evolutionary perspective, especially concerning the genetic basis of rapid adaptation to human-modified environments, as well as from a conservation perspective. I generally thought that the author’s analyses displayed ingenuity and were sound. I also enjoyed the discussion of the potential functional roles of the candidate adaptive loci within the different environmental contexts presented in the Discussion; I often feel that such discussions are quite speculative, but in this case seemed well-supported and sensible, especially given the other cited evidence. Below, I provide some ways in which I think the authors can improve this work to ensure the accuracy of their results and inferences.

(1) My first main concern relates to linking landscape features with genetic connectivity, specifically, using FST as a measure of (especially recent) gene flow. FST is a fixation index, and while it is informative about allelic differences between populations, which is affected by gene flow, FST is also affected by other demographic factors, like the effective size of populations. Furthermore, FST reflects long-term average migration rates and so while it is likely sensitive to older geographic features like rivers, FST may not be very sensitive to more recent landscape barriers or conduits to gene flow like roads, agriculture, etc., which seems important to this study. I am not saying that it is wrong to use FST to make inferences about connectivity, even on recent timescales, but it is important to consider how pairwise FST is affected by the demography, specifically, the rate of genetic drift, within the populations being compared to isolate the effects of environment on migration specifically, and also about whether FST is even sensitive enough to detect changes in migration rates arising from recent habitat modification. My suggestions are, in addition to FST as a proxy for gene flow, to examine rates of rare variant sharing between populations and apply methods that directly estimate migration rates. For example, you could consider the proportion of shared doubletons between subpopulations, that is, doubletons that are characterized by one copy of the globally rare allele occurring in one subpopulation and the other copy occurring in the other subpopulation (versus both copies of the rare allele within one of the subpopulations). Rare allele sharing of this kind would be consistent with recent migration because most globally rare variants would be relatively young and so the most likely way they are shared between populations is from recent migration. I suspect that looking at rare variant sharing would be more powerful for detecting differences in connectivity at more recent timescales. In addition, I would strongly suggest trying to use an approach like FEEMS (https://doi.org/10.7554/eLife.61927) to make direct inferences about migration. The migration surfaces produced by FEEMS would be ideal for associating with the landscape feature surfaces that the authors have generated. Also, since the authors focus on functional connectivity it could be useful to compare genome-wide estimates of genetic connectivity to estimates of gene flow at candidate adaptive loci in relation to different landscape features or environmental gradients.

(2) A second issue that I was left wondering about is how much differences in SNP sets, the distribution of linkage disequilibrium across the genomes, and differences in LD between the P10 and P20 affect inferences about the similarity/dissimilarity of the genomic architecture of environmental adaptation between the P10 and P20 toads. The density of quality controlled SNPs, even in the more liberally filtered set of SNPs, was low given the R. marina genome size (only 8,753 and 13,544 SNPs in the P10 and P20 groups, respectively), and surely adaptive signals from any truly causal adaptive variant is being detected through linkage. For interpreting overlap and differences in the genetic underpinnings of adaptation to different environments and identifying which environmental factors are relevant given the analyses that the authors performed between P10 and P20 toads, I would argue that it is important to analyze the same set of genomic sites in both groups, otherwise the discovery of candidate loci between groups is biased. In addition, because adaptive loci are likely to be detected through linkage of analyzed SNPs to casual variants (and not from the causal variants themselves) it will be important to know how the distribution of LD differs between P10 and P20 groups, so I would suggest calculating pairwise LD between SNPs within the P10 and P20 groups using Plink or ngsLD, for example, noting that population structure will influence estimates of LD and so calculations might need to be carried out within subpopulations if structure is strong. Given a common set of SNPs between groups, if LD is higher in one group due to, e.g., smaller Ne, one could be more likely to discover more candidate adaptive loci in a RAD dataset. I realize that interesting loci might be lost by paring down to a common set of quality sites shared between P10 and P20 groups, however for comparative analyses I think this is important. The authors can still present all potentially adaptive loci discovered within each of the groups given their full respective sets of SNPs (as is done now), but I think the authors should approach comparing sets of SNPs and loci between groups with caution and mindfulness of the biases that using different SNP sets involves.

In general, all inferences about neutral genetic processes (estimates of diversity, FST, migration, genome-wide structure, etc) should be based on the same set of sites for both the P10 and P20 groups. Likewise, at least for inferences involving comparisons between the P10 and P20 groups, the ‘full data’ set should consist of the same sites for P10 and P20. Otherwise, comparing estimates of population genetic parameters, summary statistics, and sets of identified adaptive loci will be biased.

I am also not sure whether “loci”, for example in lines 478–482, is being used synonymous with “SNP”. That is, do the authors mean a SNP, a region, a gene, etc? It would be helpful to readers if the authors were specific about what a locus refers to throughout the paper.

(3) My third major recommendation relates to vetting the candidate loci and to ruling out the potential of false positives (and making inferences based on them), and to generally boost confidence in the validity of the discovered candidate loci. Given that candidate loci were chosen based on statistical tests, I would recommend that the authors present Q-Q plots of the expected versus observed SNP p-values for the various environmental association tests. Providing such Q-Q plots in the supplement will help the authors and other researchers be confident that the statistical tests behaved as they should, and were not being confounded by population structure for example, which would tend to inflate signals of association across the genome and lead to the discovery of falsely significant loci. It was unclear to me whether confounding ancestry/population structure was being sufficiently accounted for in the genetic association tests with environment; presenting Q-Q plots would make things like this clear. I appreciated that the authors tested how much environmental variation was explained by randomly chosen subsets of SNPs versus the candidate loci, which does strongly suggest that population structure is not a confounding factor in the association testing, but it would still be nice to see the p-value Q-Q plots.

Also, assuming that there is sufficient density of high quality SNPs across the genome (which could likely be achieved through modified SNP filtering), I recommend that the authors test whether patterns of genetic diversity are consistent with selection acting on the candidate adaptive loci, which will help validate the candidates. The authors could likely use R. marina as an outgroup to perform HKA tests for non-neutral evolution at the candidate loci and identify what types of selection are responsible for their adaptive evolution. With sufficient SNP density, the authors could also use statistics like Tajima’s D or other local SFS-based approaches (e.g., SweepFinder) to assess whether the distribution of genetic diversity at candidate loci is consistent with selection, or at least compare levels of genetic diversity at candidate loci to genome-wide levels to determine whether within population polymorphism is low relative to genomic background levels consistent with selective sweeps, for example.

(4) It was not clear to me whether the raw FASTQ data have been made publicly available. The raw sequence data would be needed to evaluate how different filtering, mapping, and variant calling procedures affect downstream results (not just the VCF available from Dryad). Also, it would be helpful if the authors provided the data points from which summary statistics were calculated, e.g., the average depth of coverage for each sample (used to calculate the average per individual depth of coverage), heterozygosity for each individuals (used to estimate mean heterozygosity and perform statistical tests), etc. The authors have explained broadly how some quantities were calculated, e.g. heterozygosity was calculated with Stacks, however there is no mention of what inputs were used and where the exact code for carrying out analysis can be found. This is why I selected “No” for the question of whether the authors have made all data underlying the findings in their manuscript available.

Minor Comments:

83–86: Change to “Identifying which environmental variables have higher leverage on patterns of functional connectivity is therefore essential to understanding whether local adaptations evolve in isolation or in the face of gene flow and what the main extrinsic cues shaping selective pressures are.”

184: Do the authors mean “NovaSeq lanes”, not “lines”?

188: What do the authors mean by the Phred quality score of reads? That is, what aspect of read base, mapping, or alignment quality does the Phred quality of reads refer to?

195: Change “SNPs” to “SNP”.

198: It’s not clear what a minimum depth of 10 samples means. Are the authors alluding to a minimum number of samples with a minimum sequencing depth of coverage for retaining sites?

198– 99: Does maximum amount of missing data refer to a maximum number of individuals with missing genotypes?

199: What determined the minimum MAF cutoff of 10%? It would be helpful if the authors could justify why such a high MAF cutoff was used (and also what this means for inferring population structure when variation might be private to some subpopulations).

I understand that cutoffs for bioinformatic filtering can be arbitrary, but these decisions can influence estimates of population genetic parameters and effect downstream inferences, and so I believe that it is generally good practice to justify chosen cutoffs. This ensures confidence in findings. I would suggest that the authors examine the genome-wide site frequency spectrum (SFS) for each population based on their filtered set of data. I suggest to examine the SFS because the proportion of SNPs belonging to each allele frequency category should be proportional to 1/x, where x is the allele frequency. While demography can skew the shape of the genome-wide SFS, it can generally only due so so much, and excessive deviations from the expected distribution of allele frequencies can clearly point out if quality control provides a reliable set of SNPs, e.g., the SFS can inform at which point you accurately identify rare variants (so inform on MAF cutoffs), when SNP calling and allele frequency estimation is confounded by paralogy or other regions of the genome that are collapsed in a reference assembly or otherwise refractory to short read mapping. The reason why estimating the SFS accurately is so important is because many population genetic summary statistics on which the authors’ analyses rely are direct functions of the SFS. For example, estimates of genetic diversity (Watterson’s theta, nucleotide diversity, heterozygosity) are all direct functions of the SFS, while FST is a function of the joint SFS between multiple populations. Therefore, if there are clear problems in accurately estimating the SFS, which is fairly straight forward to identify given the known theoretical shape that it should approximately have, then these quantities will not be estimated accurately. Therefore, ensuring that the SFS for each subpopulation roughly aligns with the expected SFS will validate the bioinformatic processing and ensure that downstream results are reliable. If this is not the case then the authors will need to revise their quality control criteria. I recommend supplying plots of the SFS for each subpopulation and/or groups of samples with low structure (e.g. P10 and P20) calculated for sets of analyzed sites in the supplementary materials.

Also, the SFS (and hence quantities focused on by the authors) are sensitive to allelic dropout inherent to RADseq approaches, e.g. see https://doi.org/10.1111/mec.12276. It is not totally clear to me how the authors bioinformatic processing addressed this (I suspect it was how missing data was filtered, which also was not totally clear to me). This, again, is why I stress that examining the SFS for each subpopulation and/or group to ensure that it looks sane conditional on the set of quality controlled genomic sites would be a good idea.

200–201: I think the authors mean that SNPs with a p-value less than 0.05 for a test of deviation from HWE were removed. As written, the authors would be removing sites with high evidence of being in HWE, which would be undesirable. I suspect the authors accidentally switched the inequality sign and mean “p < 0.05”.

387–388: I suggest reporting standard deviation along with mean heterozygosity to provide better understanding of what the distribution of heterozygosity between P10 and P20 is.

390–394: It seems that the authors have only reported results about genome-wide levels of differentiation within landscapes, i.e., within P10 and P20. I think it would be valuable to report how diverged toads from the P10 landscape collectively are from P20 toads, that is, P10 vs P20 genome-wide FST, which will help genomically contextualize the comparisons of functional loci between P10 and P20 toads.

Similarly, I suggest that the authors perform a joint PCA and admixture analyses of all toads (P10 and P20 pooled) to ensure that genome-wide patterns of genetic relationships are in line with expectations, which will (1) help validate the bioinformatic processing that yielded the set of SNPs upon which comparisons of adaptive loci rely, and (2) help with interpretation of reported differences (and similarities) in the genetic basis of environmental adaptation.

596–597: Where the authors state “low to medium density of temporary water bodies (natural and artificial)”, I suggest the authors reiterate what low–medium means with a range of exact values, for ease of interpreting these big takeaways.

637–639: “In fact, NOTCH was discovered in the frog Xenopus, in which it is active during early embryonic stages and in the development of the nervous system [100,101]”: It’s not clear to me why the focus on NOTCH in Xenopus cited here is relevant or supports why NOTCH-related processes confer adaptation to different environments. I would argue this statement is not needed given what follows.

6. PLOS authors have the option to publish the peer review history of their article (what does this mean? ). If published, this will include your full peer review and any attached files.

**Do you want your identity to be public for this peer review?** For information about this choice, including consent withdrawal, please see our Privacy Policy .

Reviewer #1: No

Reviewer #2: No

---

## [Author Response · Author response to Decision Letter 1]

9 Jun 2025

Our response to the Editor and the reviewers comments is submitted in the editorial manager system ("Response to reviewers.pdf) as an attached file.

We've checked your submission and before we can proceed, we need you to address the following issues:

1. We note that you state the following regarding your Figures 1, 2, and 3: "Information of geostatistical boundaries was obtained from a public domain source [http://conabio.gob.mx/informacion/gis/] and the map was created by us using QGIS 3.22.10 software."

However, we note that the licensing from the above listed source is CC BY-NC 2.5, which means the content cannot be used for commercial purposes. Please note that PLOS publishes under CC BY 4.0 which allows for commercial use.

RESPONSE:

There must be a confusion. We had resolved the issue indicated regarding the source of the map in our figures 1, 2 and 3.

We downloaded the geopolitical map from:

http://geoportal.conabio.gob.mx/#!l=anfibios:1@m=topo

Which is under a CC BY 4.0 license.

We updated the web address from where we obtained the map information, which effectively is under CC BY 4.0 licensing (see pdf file submitted as 'Other' named "Map public domain source CC BY 4.0")

Figure 1. Landscape structure and sampling site location for Rhinella horribilis within the state of Oaxaca, southern Mexico. ........ "Information of geostatistical boundaries was obtained from a public domain source [http://geoportal.conabio.gob.mx/#!l=anfibios:1@m=topo] and the map was created by us using QGIS 3.22.10 software"

Thus, we proceeded with the resubmission process (with no changes since none were needed)

---

## [Decision Letter · Decision Letter 1]

18 Jul 2025

PONE-D-25-03029R1Functional connectivity patterns of the giant toad Rhinella horribilis in anthropogenically modified landscapesPLOS ONE

Dear Dr. Vázquez-Domínguez,

Thank you for submitting your manuscript to PLOS ONE. After careful consideration, we feel that it has merit but does not fully meet PLOS ONE’s publication criteria as it currently stands. Therefore, we invite you to submit a revised version of the manuscript that addresses the points raised during the review process.

The manuscript has improved a great deal, with your revisions and the reviewers guidelines. The reviewer adds a few suggestions on wording and framing, and in some cases asks for clarifications or questions interpretation. Please fix these.

The main remaining issues are the following.

Take a closer look at the landscape genetics part, see comments by reviewer 1.Provide copies of the scripts used for the analyses, like in R. This means you have to clean up the script and comment it, and ideally submit to Github with relevant datafile.

We look forward to receiving your revised manuscript.

Kind regards,

Arnar Palsson, Ph.D.

Academic Editor

PLOS ONE

Journal Requirements:

Additional Editor Comments:

The manuscript has improved a great deal, with your revisions and the reviewers guidelines. The reviewer adds a few suggestions on wording and framing, and in some cases asks for clarifications or questions interpretation. Please fix these.

The main remaining issues are the following.

1. Take a closer look at the landscape genetics part, see comments by reviewer 1.

2. Provide copies of the scripts used for the analyses, like in R. This means you have to clean up the script and comment it, and ideally submit to Github with relevant datafile.

Comments from editor

Line 54

Dont see how surviving human impact is key for long term evolution? Agree human impact is major and many organisms have to evolve to survive, but there I dont see need to talk about “long term evolution“?

Line 81-

This sentence is a bit convoluted, Here a suggestion ““Identifying environmental variables that most strongly affect functional connectivity is therefore essential to understand the level of fragmentation, whether local adaptations evolve in isolation or in the face of gene flow and what may be the main selective pressures.”

Line 105- and elsewhere

Why use the word “landscapes“ not just “areas“?

Line 130

Add info on HOW many individuals were sampled.

Line 144

Indicate that the PCA and admixture are estimated from XXX SNPs from ddRADseq.

Line 210-

Seems to lack of info here. “We applied PCA to both the study wide and the population-study datasets, while DAPC only for the latter. We ran PCA with the glpca function in adegenet v.2.1.3 in R [50], retaining 50 and 26 factors for P1O and P2O, respectively.“

PCA on whole set, how many factors were retained? And why keep different nr of factors for the two areas?

Line 326.

Maybe just skip “landscape“ and refer to P10 and P20.

Reviewers' comments:

Reviewer's Responses to Questions

**Comments to the Author**

1. If the authors have adequately addressed your comments raised in a previous round of review and you feel that this manuscript is now acceptable for publication, you may indicate that here to bypass the “Comments to the Author” section, enter your conflict of interest statement in the “Confidential to Editor” section, and submit your "Accept" recommendation.

Reviewer #1: (No Response)

2. Is the manuscript technically sound, and do the data support the conclusions?

Reviewer #1: Partly

3. Has the statistical analysis been performed appropriately and rigorously? 

Reviewer #1: Yes

4. Have the authors made all data underlying the findings in their manuscript fully available?

Reviewer #1: No

5. Is the manuscript presented in an intelligible fashion and written in standard English?

Reviewer #1: Yes

6. Review Comments to the Author

Reviewer #1: My comments to the authors are provided in an attached PDF file. They are also pasted below, but formatting may be less clear.

I was Reviewer #1 for the previous submission. I am pleased to say that the authors have greatly improved the clarity of the manuscript and rigor of data filtering, analyses, etc. I also appreciate that the removal of the section concerning genomic signatures of local adaptation may not have been an easy decision. I want to clarify that the ideas that were presented in that section were interesting, and I hope to see a future paper focused on that topic. I have some remaining feedback, including two major concerns regarding 1) the omission of relevant landcover categories in the isolation by resistance modeling; and 2) the clarity of conclusions. Feedback related to these concerns and more minor comments are detailed in my line comments below.

• L28-29: This sentence is a bit challenging to read. What about “Anthropized environments often fragment native habitats and alter the movement of individuals across the modified landscape mosaic”? More broadly, many sentences in the abstract and throughout the paper have relatively complex structures that make them challenging to read (e.g., many interjecting phrases).

• L32: This sentence can be streamlined. E.g., “We used ddRAD-seq genomic data to study the genetic diversity, genetic structure, and functional connectivity of Giant Toad (Rhinella horribilis) populations across two landscapes with distinct levels of habitat modification.”

• L30: The word “tolerance” here seems out of place since this study is focused on characterizing landscape features that affect population structure. The analyses do not directly address the extent to which the study species is tolerant to modified habitat. I think that this phrase can be omitted without changing the sentence much. E.g., “Deciphering the environmental factors associated with population genetic differentiation is essential for the conservation of wild populations inhabiting increasingly modified habitats.”

• L36: “… associated with its functional connectivity…” Replace “its” with “the toad’s” for clarity.

• L37: This sentence reads as though the differences in habitat modification are why toads in the two study areas are genetically differentiated. This is not clearly supported by the landscape genetics analyses, which primarily focused on water bodies, vegetation, and abiotic variables like solar radiation.

• L55-59: This sentence is quite long. I recommend splitting it up.

• L62: Replace “usually” with “often” or “can”.

• L64: “and even thrive” seems a bit informal and is not adding much to the sentence.

• L64: Split the sentence before “facilitated.” I.e., “Nevertheless, some species can persist, and even thrive, in modified areas. Persistence can be facilitated by specific adaptations in their life history, ecology and functional traits that counteract the negative consequences of isolation.”

• L66-67: This sentence again does not accurately describe the consequences of increased connectivity for local adaptation. Mutation is responsible for the evolution of genetic variants. Locally adaptive genetic differentiation (not locally adaptive variants) is what may not evolve when gene flow is strong enough to homogenize allele frequencies.

• L82: Replace “higher” with “high” since an explicit comparison is not being made.

• L81-84: The repeated discussion of local adaptation may not be warranted in the Introduction given that the tests for genomic signatures of spatially varying selection have been removed in this version of the manuscript. These sentences seem better suited for the Discussion section.

• L94: Remove the word “successful” and keep “common and abundant”. I.e., “One of the most common and abundant amphibian species…”

• L98-99: Remove “(i.e., detrimental conditions).” It is not necessary given the earlier contents of the sentence.

• L99: Include the average weight, or a range of weights, when stating that the species has a medium-large body size. I.e., “Its medium-large body size (X kg), terrestrial habits…”

• L102-104: It is not clear to me how these ecological characteristics make this species particularly well-suited to the study question. Do these characteristics contrast with other amphibian species in the same general geographic area? And if so, is the authors’ message here that studying the Giant Toad would thereby improve the representation of diverse ecology in research concerning anthropogenic impacts on amphibians?

• L104: Replace “anthropically” with “anthropogenically” here and throughout.

• L109-110: What life history traits led to the prediction that the two landscapes are likely genetically differentiated? Would the high dispersal capacity (L100) not make it likely that there is little genetic structure across the species’ geographic range?

• L20-123: the claim that P1O is more strongly modified than P2O is still not clear to me. In Fig1, it seems that the landscapes might have similar amounts of human-modification but different spatial configurations of human-modified land. I.e., landcover categories in P2O appear more strongly spatially autocorrelated or “patchier”. Perhaps the authors could provide a quantitative comparison of the environmental conditions in the two landscapes. E.g., what proportion of each landscape is in each landcover category?

• L195: Replace “string” with “stringent”?

• L205: It might be clearer to state the two levels of analysis as “at the total dataset level, considering all 15 localities together” and “within each of the P1O and P2O landscapes separately.”

• L214 and throughout the manuscript: “a priory” should be replaced with “a priori”.

• L232: The use of a t-test to evaluate differences in genetic summary statistics between landscapes seems inappropriate. Firstly, the authors do not state that they evaluated the normality and equal variance assumptions underlying the t-test. Secondly, the level of observation here may be the spatially distributed localities, which are not independent of one another. The authors should employ a more sophisticated analysis to test for differences between the landscapes. E.g., the authors could use the nlme package to fit a model with landscape as the focal predictor and specify the spatial autocorrelation structure with something like corGaus.

• L237-243: Please provide the pairwise correlation matrix for each landscape as a supplementary figure.

• L261-271: The manual assignment of cost/resistance values to rivers and roads in the MRDM analysis strikes me as odd because methods used later in the paper (ResistanceGA) can estimate these values objectively. ResistanceGA is capable of handling categorical data, so why not include roads and rivers in the set of models optimized with ResistanceGA? Isolation by barrier is essentially a specific sub-concept within the broader concept of isolation by resistance. Furthermore, the failure to account for roads and rivers in the ResistanceGA models could affect the inferred shapes of the relationships between other environmental predictors and landscape resistance, especially if certain values of other environmental predictors tend to co-occur with roads.

• L284-314: As stated previously, ResistanceGA can handle categorical predictors like landcover classes. Based on Fig.1, there are a few broad landcover categories present in both study areas that could affect connectivity among toad populations. It would be possible to explicitly test whether urban landcover and/or cropland are associated with high landscape resistance. Indeed, doing so would bolster the authors’ claim that anthropogenic activity affects connectivity in this species.

• L294: Remove the phrase “all combinations of these parameters.” Genetic algorithms like the one used in ResistanceGA are intended not to perform an exhaustive search of a large parameter space.

• L298: the phrase “maximize the relationship between pairwise genetic distances… and pairwise landscape distances” is not very clear. The authors should state which model fit statistic is being optimized by ResistanceGA’s genetic algorithm. The default is log-likelihood (method = “LL” in the GA.prep function).

• L378: Table 2 appears out of order based on the presentation of the results in the main text. The current Table 2 should be relabeled as Table 1.

• L381-383: How do the authors conclude that the negative relationship between roads and FST in P2O means that roads do not impact connectivity? To me it would suggest that roads facilitate or enhance connectivity in P2O. Please clarify.

• L400: “distance” meaning geographic distance?

• L457-458: The authors state “Our findings show that the landscapes studied are genetically differentiated given their distinct degree of habitat modification.” I interpret this as the authors stating that habitat modification is responsible for the genetic differentiation observed between the two landscapes. It is not clear which results support this claim.

• L466: The authors state that “R. horribilis populations are not connected”, as indicated by genetic differentiation between the two landscape study areas. FST between the two landscapes is ~0.02 (L335), which does not support this strong claim.

• L469-470: The utility of SNP heterozygosity for determining whether a population has low or high genetic diversity in an absolute sense is not clear because these estimates are based only on loci known to be polymorphic. SNP heterozygosity is only useful in a relative sense when making comparisons within a given dataset (e.g., population A has higher diversity than population B).

• L496-510: The authors state that temporary water is important for connectivity among giant toad populations. However, the results show that higher values of kernel density probability of temporary streams are associated with higher landscape resistance in P1O (Fig5a,b). Similarly, very high values of temporary water bodies are associated with higher landscape resistance in P2O. The results appear to contradict the authors’ claim. Please clarify.

• L527-529: This section is titled “Conclusions and future directions,” but it only discusses local adaptation. Local adaptation is not addressed in this manuscript, so I suggest that this section’s title is changed to only “Future directions.”

• L535-537: The authors state “Also, including populations from areas with no anthropogenic impacts, to assess if their populations exhibit genetic variants that are fixed or specifically associated to these environments but absent from natural (conserved) ones.” I think the authors mean that they could do a test for spatially varying selection between natural and anthropogenically modified habitats, but this is not clear here.

• Table 2: The model labels (“road 1”, “road 2”…) are not very informative and heavily rely on additional information found in the supplement to be interpretable. I suggest the authors make this table more self-contained.

• Fig1a: The dark red points showing the two study areas are difficult to see due to the dark green background shading. Can the red be changed to a different color? E.g., yellow-filled points with black outlines?

• Fig1d: Please include the percent variance explained by each PC axis in the axis labels. This suggestion also applies to Fig2e and Fig3e.

• Fig1e: Are the individuals ordered in any way beyond the P1O vs P2O distinction? I am curious about where the P2O individuals with relatively high P1O ancestry were sampled. If not already done, it might be interesting to order the individuals by latitude. I suspect the aforementioned P2O individuals are located at the northern portion of P2O.

• Fig2b/c: Please re-code the colors to be more consistent between SNMF and TESS. Based on the ZAA and ZAB localities, the red cluster in SNMF seems to correspond closely to the green cluster in TESS. Comparison would be easier if red and green were switched for either panel b or c.

• Fig3b/c: Please flip the orientation of the colors to be more consistent between SNMF and TESS. I.e., blue appears on top in the SNMF plot but on the bottom in the TESS plot. Comparison would be easier if red was always on top or always on bottom in both panels.

• Supplementary Methods Table S1: It is not clear that the numbers in the table are SNPs. This should be indicated in the table caption and ideally within the table itself. Similarly, it is not clear that the numbers in parentheses in the “mac” column reflect different mac thresholds. I think transposing the table might make it clearer. I.e., have one column titled “Filter”, which would contain the different filtering steps used as column names in the current table. Then add a second column titled “SNPs remaining”, which would contain the numbers of SNPs at each filtering step. The same modifications should be made to Supplementary Methods Table S2.

• Figure S6: Please ensure all the panels use the same range of values for the x- and y-axes to enable comparisons between the landscapes. Please also include a panel showing isolation by distance across the total dataset (both P1O and P2O combined).

• Finally, I have a remaining concern regarding data availability. The authors state in their response “Importantly, we did not develop any specific code for the analyses performed; we followed the scripts provided within each of the programs used, while describing in detail all the parameters applied within the manuscript” as a justification for excluding scripts from the data repository associated with this manuscript. While the authors may not have developed custom software for their analyses, they must have written their own scripts (e.g., in R) to run the analyses they cited. It is important that repositories (e.g., Dryad or GitHub) contain the scripts the authors wrote to perform analyses, as written descriptions of methods often end up being insufficient to exactly reproduce what was done. For example, the ResistanceGA analysis relies on the user writing one or more scripts with at least three different function calls and requires carefully formatted environmental and genetic data.

7. PLOS authors have the option to publish the peer review history of their article (what does this mean? ). If published, this will include your full peer review and any attached files.

**Do you want your identity to be public for this peer review?** For information about this choice, including consent withdrawal, please see our Privacy Policy .

Reviewer #1: No

---

## [Author Response · Author response to Decision Letter 2]

18 Aug 2025

Our response to the Editor and the reviewers comments is submitted in the editorial manager system ("Response to reviewers_R2.pdf) as an attached file.

---

## [Editor Report · Decision Letter 2]

15 Sep 2025

PONE-D-25-03029R2Functional connectivity patterns of the giant toad Rhinella horribilis in anthropogenically modified landscapesPLOS ONE

Dear Dr. Vázquez-Domínguez,

Thank you for submitting your manuscript to PLOS ONE. After careful consideration, we feel that it has merit but does not fully meet PLOS ONE’s publication criteria as it currently stands. Therefore, we invite you to submit a revised version of the manuscript that addresses the points raised during the review process.

All good, except the data sharing is imperfect.

The figshare link is to another study/data. 

Please update and resubmit

We look forward to receiving your revised manuscript.

Kind regards,

Arnar Palsson, Ph.D.

Academic Editor

PLOS ONE

Journal Requirements:

Additional Editor Comments:

The data sharing did not work.

The figshare link is to another study/data.

Please update and resbmit

---

## [Author Response · Author response to Decision Letter 3]

29 Sep 2025

The one comment of the reviewer was that “All good, except the data sharing is imperfect. The figshare link is to another study/data”.

We are sorry for the confusion, the FigShare link was correct, but was not available yet (it was in building process) because we were working out on how to share the huge amount of information we have from our GBS sequencing.

We finally chose to upload the data to the NCBI (BioProject PRJNA1335177) and leave in the FigShare repository the rest of the shared information (https://doi.org/10.6084/m9.figshare.29913284). Both links are now active and published (you can check them directly).

---

## [Editor Report · Decision Letter 3]

2 Oct 2025

Functional connectivity patterns of the Giant Toad Rhinella horribilis in anthropogenically modified landscapes

PONE-D-25-03029R3

Dear Dr. Vázquez-Domínguez,

We’re pleased to inform you that your manuscript has been judged scientifically suitable for publication and will be formally accepted for publication once it meets all outstanding technical requirements.

Kind regards,

Arnar Palsson, Ph.D.

Academic Editor

PLOS ONE
---

## [Editor Report · Acceptance letter]

PONE-D-25-03029R3

PLOS ONE

Dear Dr. Vázquez-Domínguez,

I'm pleased to inform you that your manuscript has been deemed suitable for publication in PLOS ONE. Congratulations! Your manuscript is now being handed over to our production team.

Kind regards,

on behalf of

Dr. Arnar Palsson

Academic Editor

PLOS ONE